Subject Areas:
evolution/palaeontology

Keywords:
*Cloudina*, Cambrian Explosion, cnidarians

Author for correspondence:
Tae-Yoon S. Park
e-mail: typark@kopri.re.kr

†These authors contributed equally.

# Enduring evolutionary embellishment of cloudinids in the Cambrian

Tae-Yoon S. Park[1,2,†], Jikhan Jung[1,2,†], Mirinae Lee[1], Sangmin Lee[3], Yong Yi Zhen[4], Hong Hua[5], Lucas V. Warren[6] and Nigel C. Hughes[7,8]

[1]Division of Earth Sciences, Korea Polar Research Institute, Incheon 21990, Republic of Korea
[2]Polar Science, University of Science and Technology, Daejeon 34113, Republic of Korea
[3]School of Earth, Atmospheric and Life Sciences, University of Wollongong, Wollongong, New South Wales 2522, Australia
[4]Geological Survey of New South Wales, W.B. Clarke Geoscience Centre, 947-953 Londonderry Road, Londonderry, New South Wales 2753, Australia
[5]Early Life Institute and State Key Laboratory of Continental Dynamics, Department of Geology, Northwest University, Xi'an 710069, People's Republic of China
[6]Department of Geology, Institute of Exact and Geosciences, São Paulo State University, 24A Avenida, Rio Claro 13506-900, Brazil
[7]Department of Earth and Planetary Sciences, University of California, Riverside, CA 92521, USA
[8]Geological Studies Unit, Indian Statistical Institute, 203 B.T. Road, Kolkata 700108, India

 T-YSP, 0000-0002-8985-930X; SL, 0000-0001-6984-0575

The Ediacaran–Cambrian transition and the following Cambrian Explosion are among the most fundamental events in the evolutionary history of animals. Understanding these events is enhanced when phylogenetic linkages can be established among animal fossils across this interval and their trait evolution monitored. Doing this is challenging because the fossil record of animal lineages that span this transition is sparse, preserved morphologies generally simple and lifestyles in the Ediacaran and Cambrian commonly quite different. Here, we identify derived characters linking some members of an enigmatic animal group, the cloudinids, which first appeared in the Late Ediacaran, to animals with cnidarian affinity from the Cambrian Series 2 and the Miaolingian. Accordingly, we present the first case of an animal lineage represented in the Ediacaran that endured and diversified successfully throughout the Cambrian Explosion by embellishing its overall robustness and structural complexity. Among other features, dichotomous branching, present in some early cloudinids, compares closely with a cnidarian asexual reproduction mode. Tracking this morphological change from Late Ediacaran to the Miaolingian provides a unique glimpse into how a primeval animal group responded during the Cambrian Explosion.

# 1. Introduction

Biotic replacement during the Ediacaran–Cambrian transition separated the Proterozoic and Phanerozoic faunas, and represents one of the most remarkable evolutionary events in the history of life [1]. It was followed by the Cambrian Explosion, during which the fossil record of various extant animal phyla began to appear, along with the Earth's first complex ecosystem engineering [2]. Following the putative extinction of the Ediacaran biota, animals showed significant increase in diversity and disparity over a relatively short interval of geological time during which origination rates greatly exceeded extinction rates [3]. The Cambrian Explosion led to defensive strategies against predation, including significant size increase [4]. However, the framework on which this novelty was built remains shrouded because Ediacaran antecedents are unknown, evidence of lineages spanning the Ediacaran–Cambrian transition and the Cambrian Explosion being scant. Establishing phylogenetic linkage across this interval can help illuminate the sequence of innovations that occurred across it and thus has considerable implications. However, securing such links is challenging for several reasons. Most Ediacaran animals and their ecologies were profoundly different from the forms of Cambrian and later animals and their life habits, which, along with the challenge of establishing secure homology in these structurally simple forms, is a reason why making phylogenetic linkages between them has proven so challenging. Putative synapomorphies exist [5] but these are subtle and overwhelmed by distinct features of life in the Late Ediacaran ecosystem. Particular attention has focused on the evolution of the first biomineralizing organisms, partly because biomineralization is a major motif of Cambrian and later life. The earliest biomineralized animals occurred in the terminal Ediacaran and had tubular construction in general. Understanding their affinities, however, is challenging because of the relative morphological simplicity of these primeval structures, which are complicated by the apparent similarity shown by more derived metazoans. For example, the branching and the 'funnel in funnel' modular construction of the latest Ediacaran tubes have parallels in many groups of annelids such as serpulids and siboglinids [6]. Recent researches have used these and other features to link the latest Ediacaran organisms to particular triploblastic groups [7,8].

The genus *Cloudina* (for original taxonomic author citations see below) is a globally occurring fossil that epitomizes tubular organisms common in the Ediacaran assemblage [9–11]. The lightly biomineralized funnel-shaped skeleton of *Cloudina* marks the earliest known usage of skeletal calcium carbonate [12]. The predatory borings found in several skeletons of *Cloudina* [13,14] demonstrate that macrophagous predation was already present at that time. Recently, cloudinids were also discovered at the basal Terreneuvian (ca 539–521 Ma) [15,16], indicating that they were also present during the ongoing biotic replacement at the Ediacaran–Cambrian boundary. Despite the fact that *Cloudina* is one of the most studied Ediacaran organisms, its phylogenetic affinity has remained debated; suggestions including being a diploblastic metazoan such as a stem- or crown-group cnidarian [6,17] or an animal with annelid/bilaterian affinity [10,18].

Based on detailed observations on new and pre-existing materials, using MicroCT, here we present character analysis which shows that cnidarian-related organisms from Cambrian Series 2 (ca 521–509 Ma) and Miaolingian (ca 509–497 Ma), mainly *Cambroctoconus*, *Lipopora* and *Tretocylichne* (the CLT clade hereafter herein) were closely related to and probably descended from cloudinids. In doing this, we acknowledge that confidence in the homology of characters from simple morphology is limited, and that the origins of several metazoan phyla may lie within the group of biomineralized tubular organisms knows as cloudinomorphs [7,8]. These caveats notwithstanding, our results reveal a phylogenetic linkage with the CLT clade based on key characters whose basis as homologies is both comparably or better justified compared with those supporting other associations and more likely, given that the CLT clade is diploblastic. If correct, this indicates that cloudinids were animals with a cnidarian affinity, which survived the Ediacaran–Cambrian boundary, participating in the Cambrian Explosion, and diversified at least until the Miaolingian. By establishing this phylogenetic connection, we then chart the pattern of character innovation across this critical interval.

# 2. Material and methods

This study is grounded in detailed morphological comparison of different species of Ediacaran and Cambrian cloudinids collected worldwide. The specimens of *Cloudina hartmannae* were collected from the Lijiagou section in Shaanxi Province, China. Fossils of *Cambroctoconus orientalis* for this study are from the Miaolingian Zhangxia (Changhia) Formation at Xintai section (35°45′05″ N, 117°46′22″ E) in Shandong

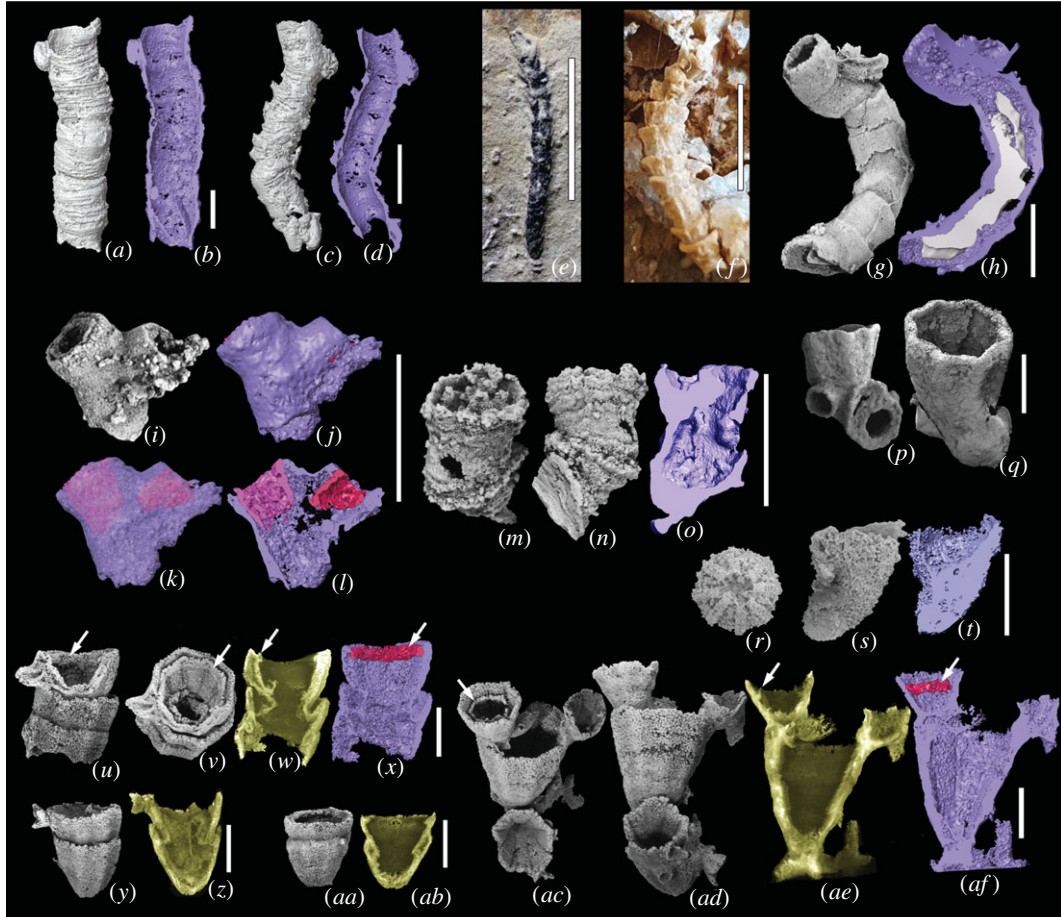

**Figure 1.** Images of cloudinids from Late Ediacaran to Cambrian with their MicroCT-rendered images. (*a–d*) *Cloudina hartmannae* from the terminal Ediacaran of China. (*a,b*) MicroCT-rendered images of KOPRIF5501; (*a*) lateral view; (*b*) longitudinal-cut view. (*c,d*) MicroCT-rendered images of KOPRIF5502; (*c*) lateral view; (*d*) longitudinal-cut view. (*e*) *Cloudina lucianoi* collected from the uppermost Ediacaran to the basal Cambrian of the Tagatiya Formation, Itapucumi Group, Paraguay. (*f*) *Cloudina carinata* from the lowermost Cambrian of the Membrillar olistostrome, central Spain, holotype, UEXP709Me2:006. (*g–l*) *Lipopora lissa* from the Cambrian Stage 4 of Australia. (*g,h*) Holotype, ANU59521; (*g*) lateral view; (*h*) MicroCT-rendered longitudinal-cut view (silica infilling is represented by greyish colour). (*i–l*) Paratype, ANU59523 J; (*i*) dorso-lateral view; (*j*) MicroCT-rendered lateral view; (*k*) MicroCT-rendered transparent lateral view showing the position of two daughter funnels formed by dichotomous branching (reddish colour). (*l*) MicroCT-rendered longitudinal-cut view. (*m–o*) *Lipopora daseia* from the Cambrian Stage 4 of Australia, holotype, ANU29553 J; (*m*) dorso-lateral view; (*n*) lateral view; (*o*) MicroCT-rendered longitudinal-cut view. (*p,q*) *Tretocylichne perplexa* from the Wuliuan of Australia. (*p*) Holotype, MU52711. (*q*) Paratype, MU52717. (*r–t*) *Cambroctoconus koori* from the Cambrian Stage 4 to Wuliuan Stage of North Greenland. (*r*) KOPRIF15002, ventral view. (*s,t*) KOPRIF15003; (*s*) lateral view; (*t*) MicroCT-rendered longitudinal-cut view. (*u–af*) *Cambroctoconus orientalis* from the Drumian Stage of China. (*u–x*) KOPRIF17008, a specimen showing two consecutively stacked funnel-shaped elements and a band-like structure (white arrows) which represents the initial growth of a new funnel; (*u*) lateral view; (*v*) dorsal view; (*w*) MicroCT-rendered longitudinal-cut view; (*x*) MicroCT-rendered longitudinal-cut view with the band-like structure represented by reddish colour. (*y,z*) KOPRIF5001, a specimen with a funnel-shaped element; (*y*) lateral view; (*z*) MicroCT-rendered longitudinal-cut view. (*aa,ab*) KOPRIF5002, a specimen with a funnel-shaped element fused with the rim of the parental cup; (*aa*) lateral view; (*ab*) MicroCT-rendered longitudinal-cut view. (*ac–af*) KOPRIF5003, a specimen with an offset at the top possessing a band-like structure (white arrows); (*ac*) oblique dorsal view; (*ad*) lateral view; (*ae*) MicroCT-rendered longitudinal-cut view. (*af*) MicroCT-rendered longitudinal-cut view with the band-like structure represented by reddish colour. Scale bars for (*a–d*) are 500 µm, for (*e–o,u–af*) are 5 mm, and for (*p–t*) are 1 mm.

Province [19]. Hundreds of silicified specimens of *Ca. orientalis* were collected, some of which were previously illustrated [19]. Specimens of *Cambroctoconus coreaensis* used in this study were previously described [20]. Three new specimens of *Cambroctoconus koori* were collected from North Greenland (see [21]). All the specimens of *Cl. hartmannae*, *Ca. orientalis*, *Ca. coreaensis* and *Ca. koori* are held in the Korea Polar Research Institute and given registered numbers prefixed with KOPRIF. Specimens of *Cloudina lucianoi* were recovered from the upper succession of the Late Ediacaran–Early Cambrian Tagatiya Guazu Formation (Itapucumi Group), Paraguay and were previously described [11,22] (figure 1*c*; electronic supplementary

material, figure S1c–e). Specimens of *Lipopora* are deposited in the Department of Geology of the Australian National University, Canberra. All the type materials including the holotype and paratypes were analysed for this study. Specimens of *Cambroctoconus* and *Lipopora* were then coated with magnesium oxide and photographed by Canon EOS 60D, while images of the specimens of *Ca. koori* were taken by scanning electron microscope (JEOL JSM-6610) equipped at Korea Polar Research Institute. Those of *Tretocylichne perplexa* are housed in Macquarie University, and were imaged by a scanning electron microscope equipped at Australian Museum.

Specimens of *Cl. hartmannae*, *Ca. orientalis*, *Ca. koori*, *Ca. coreaensis*, *Lipopora lissa* and *L. daseia* were analysed by X-ray microcomputed tomographic scanning, using the SkyScan 1172 system at the Dental Research Institute of Seoul National University, and the customized equipment housed in the Korea Institute of Geoscience and Mineral Resources (KIGAM). The scan parameters for each specimen varied depending on the size, shape and density of the specimen. Typical scans were acquired at 70 kV with a pixel size of 7.98 µm, 0.4° rotation per step.

A parsimony test was performed by using TNT v. 1.5 [23] under the traditional search with 10 random seeds, 100 replications of a random addition sequences, tree-bisection reconnection (TBR) branch swapping algorithm, saving 10 trees per replication. Sixteen ingroup taxa were included in the analysis, for which 23 characters were coded (see Character List and Character Matrix in the electronic supplementary material). *Sinotubulites* was selected as an outgroup, since it is a tubular fossil that occurred at latest Ediacaran, but lacked the typical funnel-in-funnel structure of cloudinids. All characters were equally weighted and unordered. Inapplicable and unknown character states were coded as '-', and '?', respectively. Polymorphic characters were coded as [01]. Additional phylogenetic analyses were performed for the same taxa and characters using both Bayesian and maximum likelihood methods. Maximum likelihood analysis was done in W-IQ-Tree online version [24]. The data matrix was also used for MrBayes v. 3.2.7, a program for phylogenetic inference using Bayesian statistics. The dataset was run on 5 000 000 generations and 0.1 burn infraction with lset rates as gamma.

In order to illustrate the morphological modifications detected in the cloudinids, we made a three-dimensional (3D) model for each cloudinid species considered in this analysis and other 'cloudinomorphs' from Late Ediacaran and the first three Cambrian series, and conducted quantitative measurements. Volumetric reconstructions of scans were done in NRecon v. 1.6.9.3 software. Resulting image stacks were processed and segmented in 3D Slicer v. 4.10.1 and segmentation results were exported as .STL files, which then were imported into Blender v. 2.80 for cross-sectional views and 3D rendering. Some of the 3D reconstructions (the longitudinal-cut views of yellowish colour) for *Ca. orientalis* in figure 1 were made with a resolution of 12.97 µm per voxel using the SkyScan software CTVox32. For modelling, apertural/basal widths and height of cloudinid units were measured from published images that were selected to represent each cloudinid species. These numbers were used as the basic parameters of the funnel-shaped 3D model for the species. The volume of each cloudinid unit was measured using the volume statistics function of Print3D add-on. Once the basic shape was determined, the model was stacked to simulate funnel-in-funnel structure of cloudinids. To ensure the resulting 3D models appropriately reflect the shapes of reported cloudinid fossils, cross-sections were made as needed and compared with published images. For those species that could not be analysed by MicroCT, measurements were made directly from illustrated images previously published in the literature (see tables in the electronic supplementary material). To assess the changes in volumes and the ratios of wall thickness to apertural width of cloudinids through time, Mann–Whitney *U*-tests were done using Microsoft Excel (electronic supplementary material, table S3). For the measurements, a total of 28 specimens were used, nine for Ediacaran, four for Terreneuvian, six for the Series 2, nine for the Miaolingian.

# 3. Results

## 3.1. General morphology of *Cloudina*

The morphology of *Cloudina* is characterized by the stacking of gradually expanding funnel-shaped modular elements (figure 1a–d) such that elements wider at the apertural end originate within the internal cavity of the preceding element. In the type species, *Cl. hartmannae,* there are transverse annulations on the outer surface of the funnel-shaped elements (figure 1a,c). Each funnel-like element is nested deeply within the preceding element, and they frequently flare aperturally to form flanges that may fuse to neighbouring flanges [17,18]. Dichotomous branching commonly occurs deep within

the parental funnel forming two daughter funnels of equal size, but recently, budding from the outside of the parental funnel has also been reported [25]. In *Cloudina riemkeae* the funnel-like elements are small and nested only shallowly [17]. The type specimen of *Cloudina carinata* were originally thought to occur in the terminal Ediacaran [26], but is now known to be basal Terreneuvian in age [27]. A particular morphological feature of *Cl. carinata* is its roughly eight-sided appearance assumed by longitudinal ridges on the outer surface (figure 1*f*). It is also notable that some specimens of *Cl. lucianoi* show partial angularities in transverse sections of the funnel elements [28].

## 3.2. Funnel-in-funnel-shaped skeletons of the CLT clade

*Lipopora* from the Cambrian Stage 4 of New South Wales, Australia, includes two species, *L. lissa* (type species) and *L. daseia*. The genus was described as colonial, but commonly appears as solitary individuals [29]. The individual element has a holdfast at the most basal element. While *L. lissa* has 16 thin, short septa and smooth external wall (electronic supplementary material, figure S2*r,s*), *L. daseia* shows eight broad and short septa (electronic supplementary material, figure S2*c*) and narrow annulations on the external wall (electronic supplementary material, figure S2*i,j*). The skeleton of *Lipopora* is eight-sided in some, if not all, specimens (electronic supplementary material, figure S2*g,h*). *Lipopora lissa* presents a mode of asexual reproduction by dichotomous branching (figure 1*i–l*), as well as budding from the outer surface (figure 1*g,h*; electronic supplementary material, figure S2*x–z*). A longitudinal MicroCT section image reveals that the individual element of a colonial specimen is probably of a funnel shape, assuming a funnel-stacking pattern (figure 1*g,h*). In the specimen showing dichotomous branching, the two daughter elements also form funnel-like structures (figure 1*k,l*).

The reproductive mode of *Ca. orientalis*, the type species of the genus, was characterized by the budding of new elements from the outer surface, inner surface and the rim of the parental cup [19]. Detailed observation, both of the new materials and the previously documented specimens of *Ca. orientalis* [19] (figure 1*u–af*), reveals an additional growth process that is distinct from budding. This begins with the appearance of a looped, band-like structure on the inner surface of the cup (figure 1*u–x, ac–af*), which represents an initial growth of a funnel-shaped skeleton. The funnel-shaped skeleton is sometimes fused to the rim of the parental cup (figure 1*aa,ab*). Possible funnel-in-funnel structures may be also present in *Ca. koori* ([21], fig. 3-2).

*Tretocylichne perplexa* was documented from the Wuliuan Stage (Miaolingian Series) of New South Wales, Australia [30]. It has a distinctly defined eight-sided skeleton with eight pairs of septa. The skeleton shows an open aperture at the base, so that the overall morphology forms a funnel-like structure with a holdfast and some degree of curvature (electronic supplementary material, figure S3). Budding from the outside of the parental funnel is represented by the holes in the skeletons (figure 1*p*; electronic supplementary material, figure S3*a–d,j,k*).

## 3.3. Comparison between *Cloudina* and the CLT clade

There are critical similarities between the CLT clade and the specimens that indisputably are ascribed to *Cloudina*. The production of funnel-shaped elements seen in *Ca. orientalis* and *L. lissa* is directly comparable to the funnel-shaped element of the type species of *Cloudina* (figure 1). For example, the stacking of funnel-shaped elements in *L. lissa* is almost identical in appearance to that in *Cloudina* (figure 1*g*), and even the cup-shaped elements of *Ca. orientalis* sometimes produced stacked funnel-in-funnel morphology (figure 1*w,x*). The eight-sided shape of the CLT clade skeleton is reminiscent of the roughly eight-sided appearance of the funnel element of *Cl. carinata*. Both the CLT clade and *Cl. hartmannae* show budding from the outside of the parental elements. *Lipopora lissa* and *Cloudina* show dichotomous branching as a mode of asexual reproduction, in which the diameter of the daughter element is significantly smaller than that of the parental element [18]. This was most likely formed by longitudinal fission of the parent element, an asexual reproduction mode only known in cnidarians [31]. It is noteworthy that such dichotomous branching has not been observed in other 'cloudinomorphs,' but to date only in *Cloudina* and *Lipopora*. The presence of transverse annulations in the outer surface of *L. daseia* is comparable to those in many *Cloudina* species (figure 1*m,n*; electronic supplementary material, figure S2*i,j*). *Cloudina* skeletons show evidence of being to an extent malleable [17,18], which is similar to those observed in the skeletal wall of *Cambroctoconus* interpreted as highly integrated with soft tissue [19,32]. Given these striking similarities, we consider that the Cambrian CLT clade was closely related to *Cloudina sensu stricto*.

In addition to the morphological similarity, the ecological habitats were also comparable among these organisms. *Cloudina* is interpreted as an epibenthic 'mat-sticker' on soft substrates with partial anchoring [11]. It was also suggested that *Cloudina* tubes were oriented horizontally on a soft substrate [33,34], which is reminiscent of a *Cambroctoconus* colony growing horizontally on a substrate [19]. As for *Cloudina* [10,11,33,35], *Ca. orientalis* has frequently been associated with biostromes or bioherms [19,32], suggesting possibly similar ecological habitats throughout the Late Ediacaran and Cambrian.

## 3.4. Family Cloudinidae and included species

*Cambroctoconus* was originally considered as a stem-group Cnidaria [19]. Subsequently, it was revealed that *Tretocylichne* and *Lipopora* were closely related to *Cambroctoconus* [20], hence forming the CLT clade herein. By contrast, based on a hypothesis that *Ca. koori* has a close affinity to octocorals, the CLT clade was accommodated within a new order Cambroctoconida [21] which is questionably assigned to Octocorallia [36]. Here it suffices simply to acknowledge the CLT clade was related to cnidarians. As the CLT clade was closely related to *Cloudina* it is reasonable to accommodate the CLT clade within the family Cloudinidae [37]. Eleven cloudinid species are considered in this study (see electronic supplementary material, supplementary text for other cloudinids and 'cloudinomorphs' that are excluded in this present analysis). They are:

*Cloudina hartmannae* [10]: Uppermost Ediacaran to basal Fortunian Stage, occurring worldwide. This is the type species of the genus *Cloudina* (figure 1*a*–*d*).

*Cloudina riemkeae* [10]: Nama Group (uppermost Ediacaran), Namibia. This species has frequently been regarded as a junior synonym of *Cl. hartmannae*. Given the morphology of the holotype, however, the funnel element of *Cl. riemkeae* overlaps shallowly to the parental element (electronic supplementary material, figure S8), whereas the funnel element of *Cl. hartmannae* overlaps deeply (electronic supplementary material, figure S8). Accordingly, we treat them as distinct taxa.

*Cloudina carinata* [26]: Membrillar olistostrome (basal Fortunian Stage), Badajoz Province, central Spain (figure 1*f*; electronic supplementary material, figure S1*a*,*b*). This species is characterized by having longitudinal ridges on the outer surface of elements, which give a roughly octaradial symmetry in dorsal view ([26], fig. 7A). The age of Membrillar olistostrome was originally described as Late Ediacaran [26], but is considered as the earliest Terreneuvian in the most recent study [27]. A smaller specimen was also reported from the terminal Ediacaran interval of the Villarta Formation of the Ibor Group, central Spain [38], but we believe that the smaller size makes it distinct from the types of this species from the Membrillar olistostrome. There are two specimens of *Cl. carinata* reported from the Tamengo Formation, Brazil [39]. Although slightly larger, they are similar in morphology to the types of this species from the Membrillar olistostrome (electronic supplementary material, figure S8). Considering that the precise age interval of the fossil-bearing horizon within the Tamengo Formation is relatively poorly constrained and that the measurement of the thickness in the Brazilian specimens is hampered by the very small number of specimens, they are not considered in the present analysis.

*Cloudina lucianoi* [40]: Upper Ediacaran to basal Fortunian Stage of the Tagatiya Guazu Formation, Itapucumi Group, Paraguay (electronic supplementary material, figure S1*c*–*e*), and the Tamengo Formation of the Corumbá Group, Brazil. In Paraguay, this species was originally described to occur from the Late Ediacaran [11], but the uppermost fossil-occurring interval possible reaches the earliest Fortunian. Recent U-Pb Shrimp dating performed in a volcanic ash tuff located about 15 m above the uppermost *Cloudina*-bearing bed indicates depositional age of $545 \pm 5.65$ Ma [41]. However, the ongoing re-evaluation of these data possibly indicates younger depositional ages, which will allow placing this level already in the Early Cambrian. Similarly to the Paraguayan occurrences, the age of the specimens from the Tamengo Formation is interpreted as latest Ediacaran [28]; however, they also may be younger [42], as originally considered [40]. From the point of view of their general morphology, both specimens from Brazil and Paraguay are centimetric sinuous specimens constituted by nested and regularly spaced collars. Occasionally, some better-preserved specimens show rounded closed basal end (specially the samples from Tagatiya Guazu Formation). Detailed discussions about the age, taphonomy and taxonomy of *Cl. lucianoi* specimens from the Tagatiya Guazu Formation and the Tamengo Formation were given by previous studies [11,22,28]. Awaiting precise stratigraphic data, this study excludes measurement data from *Cl. lucianoi* specimens of the Tamengo Formation, Brazil [28], only considering the specimens from the uppermost Ediacaran to the basal Fortunian of the Itapucumi Group, Paraguay. As a result, it is inferred that the stratigraphic occurrence of *Cl. lucianoi* is similar to that of *Cl. hartmannae* (figure 2).

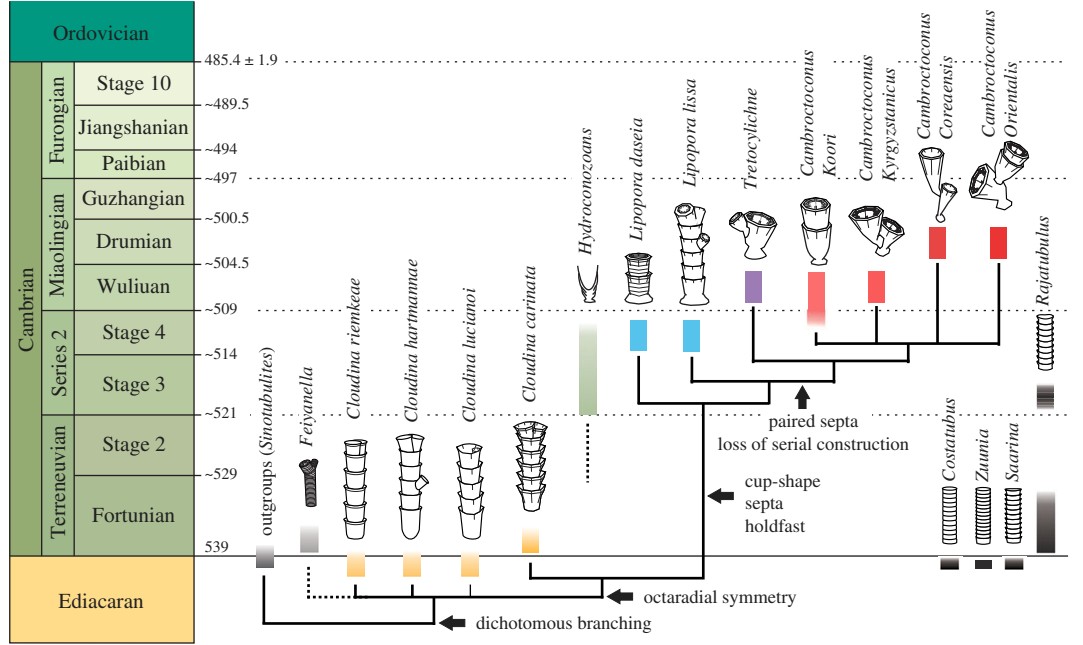

**Figure 2.** A strict consensus tree of cloudinids with their phylogenetic occurrences and diagrammatic representations of morphology. This tree is made out of the four most parsimonious trees from a phylogenetic analysis (electronic supplementary material, figure S4). The Cambrian Series 2 to Miaolingian CLT clade (see text) is distinct from *Cloudina* species, with *Cloudina carinata* being its sister group. The cloudinid-like tubular organisms, *Zuunia*, *Rajatubulus*, *Costatubus* and *Saarina* are not clustered with the typical cloudinids in the strict consensus tree. *Feiyanella* is grouped with typical cloudinids by the presence of dichotomous branching, but see the main text for further information. Bars indicate stratigraphic ranges and colours indicate generic affiliation.

*Cambroctoconus orientalis* [19]: Zhangxia Formation (Drumian Stage) of Shandong Province, China (figure 1*u–af*); type species of the genus *Cambroctoconus*. These fossils commonly show a conical cup-shaped morphology, but funnel-shaped elements are not rare. Budding from the outer surface of the funnel-shaped element is also present (figure 1*u,v*).

*Cambroctoconus kyrgyzstanicus* Peel [43]: Wuliuan Stage of Sauk Tanga, Alay range, western Kyrgyzstan [43]. Although septa were not described to be paired in the original description, given the location of septa ([43], fig. 2A and F), it is likely that they actually were paired, as in other *Cambroctoconus* species.

*Cambroctoconus coreaensis* [20]: Daegi Formation (Drumian Stage) of the Taebaeksan Basin, Republic of Korea (electronic supplementary material, figure S10). The length of the element in this species is the longest of all cloudinids (figure 3). Due to poor preservations and frequent silica infillings, not much is known for this species. Nevertheless, the presence of rejuvenation in one specimen is observed in the longitudinal-cut view (electronic supplementary material, figure S10*f*).

*Cambroctoconus koori* [21]: Henson Gletscher Formation (Cambrian Stage 4 to Wuliuan Stage), North Greenland (electronic supplementary material, figure S11). All the specimens are preserved as internal moulds [21] (electronic supplementary material, figure S11), but the rejuvenation of an element ([21], fig. 3-2) and eight pairs of septa [21] (figure 1*r*) are recognizable. In MicroCT-rendered longitudinal-cut views, the basal part appears to be thicker than the lateral walls (electronic supplementary material, figure S11). Since specimens are preserved as internal moulds, measuring of the wall thickness is not possible. The thick basal part of the internal moulds is comparable to the basal cavity seen in *Ca. orientalis* [20].

*Lipopora lissa* [29]: Coonigan Formation (Cambrian Stage 4), Mootwingee district, New South Wales, Australia (figure 1*g–l*; electronic supplementary material, figure S2Q–Z). This is the only post-Terreneuvian cloudinid showing the serial construction as Ediacaran–Terreneuvian *Cloudina* species do (figure 1*e,f* and electronic supplementary material, figure S8).

*Lipopora daseia* [29]: Coonigan Formation (Cambrian Stage 4), Mootwingee district, New South Wales, Australia (figure 1*m–o*; electronic supplementary material, figure S2*a–p*). This is the only post-Terreneuvian cloudinid showing transverse annulations on the outer surface (figure 1*m,n*; electronic supplementary material, figure S2*i,j*) as some of the Ediacaran–Terreneuvian *Cloudina* species also do.

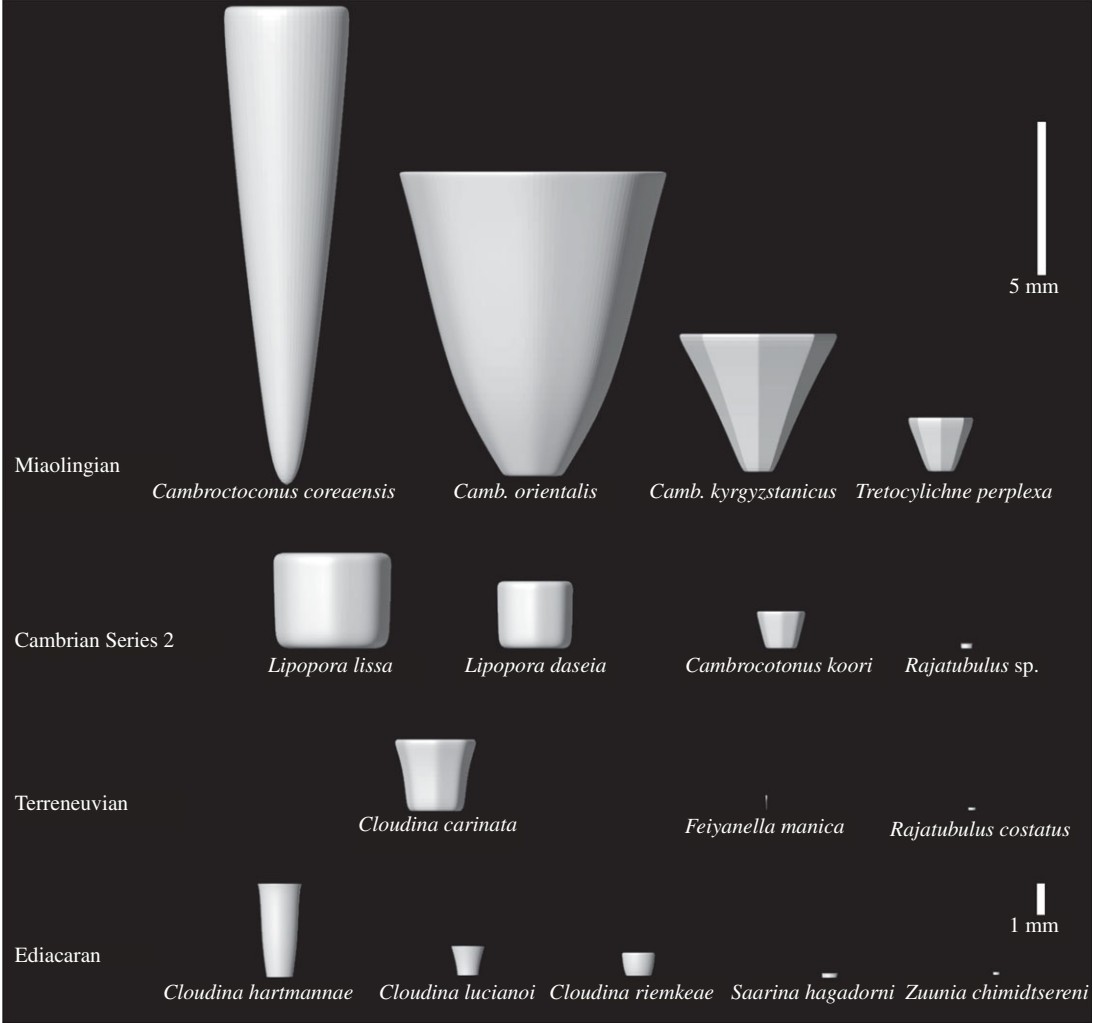

Miaolingian

*Cambroctoconus coreaensis*    *Camb. orientalis*    *Camb. kyrgyzstanicus*    *Tretocylichne perplexa*

Cambrian Series 2

*Lipopora lissa*    *Lipopora daseia*    *Cambrocotonus koori*    *Rajatubulus* sp.

Terreneuvian

*Cloudina carinata*    *Feiyanella manica*    *Rajatubulus costatus*

Ediacaran

*Cloudina hartmannae*    *Cloudina lucianoi*    *Cloudina riemkeae*    *Saarina hagadorni*    *Zuunia chimidtsereni*

5 mm

1 mm

**Figure 3.** Elemental size of cloudinids and the cloudinid-like tubular organisms at a glance. There is a tendency of size increase through time in cloudinids, probably an expression of the Cambrian Explosion. The 'cloudinomorphs,' *Zuunia*, *Saarina* and *Rajatubulus* are distinguished from cloudinids not only by their sizes, but also by their elemental morphology and absence of branching. Note that the data for *Cl. lucianoi* were acquired from an interval across the Ediacaran/Cambrian boundary (see the main text).

*Tretocylichne perplexa* [30]: Murrawong Creek Formation (Wuliuan Stage), New South Wales, Australia (figure 1*p,q*; electronic supplementary material, figure S3). The lower base of the element is consistently open, so that structurally it is more similar to funnel shape than cup shape. Of the CLT clade, this is the only species that does not show cup shape.

## 3.5. Phylogenetic analysis and morphological embellishment during Cambrian Explosion

In order to see how the morphological characters of cloudinids are distributed in phylogenetic context, we performed phylogenetic analyses with four species of *Cloudina* and seven species of the CLT clade, as well as other 'cloudinomorphs' occurring around the Ediacaran/Cambrian boundary. Parsimony test produced four most parsimonious trees with 42 tree lengths (CI of 0.595, RI of 0.770) (electronic supplementary material, figure S4), whose consensus tree shows a separate grouping of cloudinids (*Cloudina* species and the CLT clade) and *Feiyanella*, from other 'cloudinomorph' species, based on the presence of dichotomous branching (figure 2). Within cloudinids, *Cloudina* species from the Late Ediacaran and the basal Terreneuvian appear to form a stem group to the Cambrian Series 2 and Miaolingian CLT clade (figure 2); *Cl. carinata* is recognized as sister taxon of the CLT clade by having an octaradial symmetry. In the result of the Bayesian analysis, the lower cloudinids were not differentiated from the cloudinid-like tubular organisms (electronic supplementary material,

figure S5*a*), while the result from the maximum likelihood approach supports a grouping of cloudinids (electronic supplementary material, figure S5*b*).

Angularity in transverse section is frequently seen in *Cl. lucianoi*, but the transition towards regular octaradial symmetry appeared near the base of the Cambrian in *Cl. carinata* (figure 2). Since the Cambrian Explosion occurred before the pre-Miaolingian [3,44], the evolution of characteristics defining the CLT clade are here interpreted as novel strategies for surviving a progressive increase in predation pressure; through the Terreneuvian and the Early Cambrian Epoch 2, cloudinids evolved septa, cup-shaped morphology and probably a holdfast for better attachment. The largely Miaolingian *Cambroctoconus* and *Tretocylichne* are characterized by clear eight-sided shape, paired septa and the absence of serial construction but retain the funnel-in-funnel construction and other characters listed above that attest to their cloudinid origin (figure 2). Members of *Cloudina* itself are not yet known from the Cambrian stages 2 and 3 (figure 2), but there are some potential close relatives from this interval. Hydroconozoans are known as coralomorphs occurring from the Cambrian Stage 3 to early part of Cambrian Stage 4 [45] (figure 2). Although their morphology is poorly understood, being known only from thin sections, the cup-shape and tentative modular structures suggest that hydoroconozoans were related to cloudinids. The cryptic habit of hydroconozoans [45] is also comparable to that of *Ca. orientalis* [19,32]. Thus, apparently the cloudinid lineage began with a simple, round, funnel-in-funnel construction, and then acquired octaradial symmetry during the Ediacaran–Cambrian transition. Subsequently, the clade evolved cup shape, septa and the holdfast and, finally, paired septa. The funnel-in-funnel construction was progressively diminished, with cup-budding becoming the main constructional strategy. This pathway of morphological evolution connects the terminal Ediacaran *Cloudina* to the Miaolingian *Cambroctoconus*.

# 4. Discussion

The presence of octaradial symmetry and septa, and the reproduction mode of budding has placed the CLT clade at a stem-group cnidarian position [20], but an octocoral affinity for *Cambroctoconus* has also been suggested [21]. While soft-bodied evidence is required for precise determination, both interpretations affirm the cnidarian affinity for this group. Therefore, the striking morphological similarity between the CLT clade and cloudinids evince a cnidarian affinity of cloudinids [6,17]. The characteristic dichotomous branching present in *Cloudina* and *Lipopora* also corroborates the cnidarian affinity of cloudinids. This is consistent with the recently suggested non-bilaterian affinity of the cloudinomorphs from Namibia [34]. Accordingly, we conclude that typical cloudinids are the first convincing cnidarian-grade animals observed in the geologic record. Those 'cloudinomorph' animals with a 'through' gut [8] were apparently among other Late Ediacaran tube-dwelling organisms, but the absence of the funnel-in-funnel shape in these forms may differentiate them from cloudinids. Cloudinid-like tubular organisms such as *Zuunia* [7], *Rajatubulus* [15], *Costatubus* [46] and *Saarina* [47] can be distinguished from typical cloudinids by the shape and spacing of their repetitive elements (figure 3; electronic supplementary material, figure S6); their elements are notably stubby compared with those of *Cloudina* and close relatives, and show no overlapping of elements in longitudinal section images (figure 3; electronic supplementary material, figure S8). In addition, asexual reproduction including dichotomous branching shown in cloudinids, has not been documented from these genera. The elemental size of *Zuunia* and *Rajatubulus* are also significantly smaller than those of typical cloudinids (figure 3). In this study, *Zuunia*, *Rajatubulus*, *Costatubus* and *Saarina* are thus considered to be 'cloudinomorphs' distinguished from the true cloudinids. Recently, a bilaterian affinity was proposed for cloudinomorphs [7,8], which, if correct, suggests that their apparent similarity to typical cloudinids was a result of morphological convergence.

The cnidarian affinity of cloudinids in this study, however, leads us to propose an alternative hypothesis for the phylogenetic affinity of the cloudinid-like tubular organisms. Peridermal tubes of coronate scyphozoan polyps are chitinous exoskeleton which contains and protects the whole polypoid body [48]. The length of the tubes is from a few millimetres up to 1–2 cm [49], while the diameter of the tube is seldom over 1 mm at the aperture (electronic supplementary material, figure S7), being comparable to the size of the diminutive 'cloudinomorphs'. Prominent transverse rings and less prominent longitudinal striations are present on the outer surface of tubes [50]. Due to superficial similarities, they are often mistaken for polychaete tubes [51]. The soft polyps are highly elongate, and thus have a lengthy gastrovascular cavity that can retract deeply inside the tubes. Although many are solitary, some colonial species present asexual reproduction via branching or

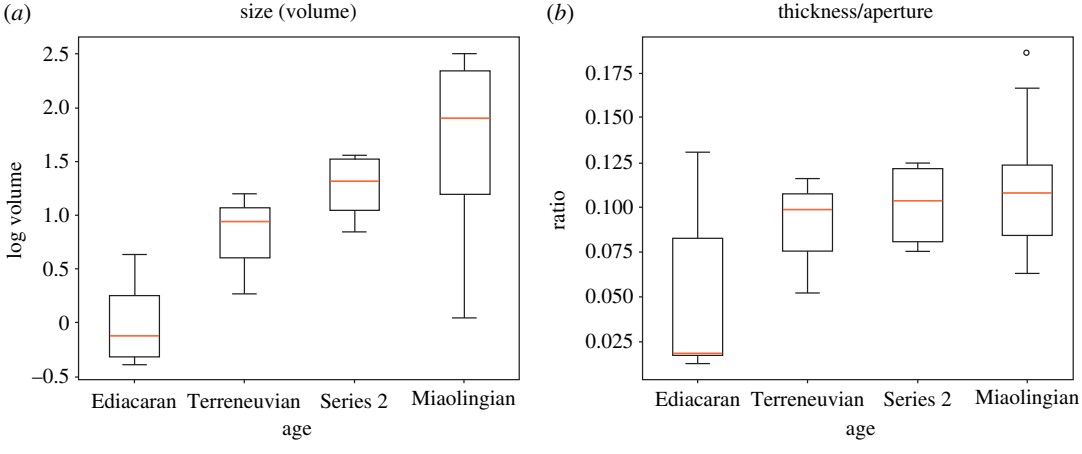

**Figure 4.** Elemental size and the wall thickness/aperture ratio of cloudinids from the Ediacaran and the first three Cambrian series. There is a progressive increase in size of each element from the Ediacaran to the Miaolingian (*a*). The dramatic increase in the elemental volume between the Ediacaran and the Terreneuvian may reflect the increase in the predation pressure near the base of the Cambrian. A generally increasing trend shown in the wall thickness/aperture ratio (*b*) indicates the relative wall thickness also increased, probably in response to the continuous increase in the predation pressure at the Cambrian.

budding [49,52,53]. Due partially to the presence of tetraradially or octoradially arranged cusps, these forms have been compared to conulariids and *Corumbella* [54,55], but the cusps are intermittently present at the inside of the tubes; and there are also species without cusps [56], which assume a similar shape and morphology to the tubes of the 'cloudinomorphs'.

The skeletal fauna of the terminal Ediacaran Nama Assemblage represents the first appearance of external hard parts in macroorganisms [57]. Until recently, typical Ediacaran organisms, both skeletal and soft-bodied, were thought to have become completely extinct, being replaced by distinctly different Early Cambrian fauna. Some researchers have proposed a mass extinction at this boundary [9], but growing evidence suggests a transitional replacement [58–60], at least regarding the occurrences of cloudinids from the Early Terreneuvian [15,16]. In either case, a notable biotic replacement near the Ediacaran–Cambrian boundary separates the Proterozoic and Phanerozoic faunas [1]. However, as demonstrated here, the recognition of the Cambrian Series 2 and Miaolingian CLT clade as cloudinids indicates that the lineage not only survived this replacement, but also endured the Cambrian Explosion which occurred mainly during the Terreneuvian and the Cambrian Epoch 2 [3,44], making cloudinids the first particular animal group of Ediacaran origin demonstrated to have diversified successfully in the Cambrian. Moreover, the evolution of novel features, such as holdfast, cup-shaped morphology, and septa, in the Cambrian Series 2 and Miaolingian CLT clade appear to have been structural innovations that helped cloudinids prosper during the Cambrian Explosion.

The morphological modifications through time are represented in the 3D model for each cloudinid species (figure 3; electronic supplementary material, figures S6 and S7) and the quantitative measurements on cloudinids from Late Ediacaran and the first three Cambrian series (figure 4). Overall, there was a progressive increase in the size of each element (funnel-shaped or cup-shaped) from the Ediacaran to the Miaolingian (figures 3 and 4*a*), but a marked increase in size is observed between the Ediacaran and the Terreneuvian, which is followed by a less prominent increase between the Terreneuvian and the Cambrian Series 2 (figure 4*a*). This result might indicate a dramatic morphological change in the cloudinid lineage in response to the increase in predation pressure occurred near the base of the Cambrian. This is in line with the notable increase in the wall thickness/aperture ratio between the Ediacaran and the Terreneuvian (figure 4*b*); the marked thickening of wall probably resisted increasing predation pressure after the start of the Cambrian. Furthermore, there is a notable increase in size from the Cambrian Series 2 to the Miaolingian (figure 4*a*). Together with the fact that the Miaolingian-occurring *Tretocylichne* and *Cambroctoconus* evolved novel characteristics mentioned above, the size increase in the Miaolingian may indicate that the premium on major structural innovation continued until the Late Cambrian Series 2.

Data accessibility. Data available from the Dryad Digital Repository: https://doi.org/10.5061/dryad.ffbg79cv8 [61]. The data are provided in electronic supplementary material [62].

Authors' contributions. T.-Y.S.P. conceived the idea and designed the project; J.J. and T.-Y.S.P. handled the MicroCT analyses with help from S.L., Y.Y.Z. and H.H. J.J. processed 3D imaging, and acquired measurement data. M.L. performed the phylogenetic analysis. T.-Y.S.P drafted the manuscript with inputs from L.V.W., N.C.H. and all other authors.

Competing interests. We declare we have no competing interests.

Funding. This research was financially supported by Korea Polar Research Institute (KOPRI) project PE21060. Y.Y.Z. publishes with the permission of the Executive Director, Geological Survey of New South Wales. L.V.W. is a fellow of the CNPq. The samples from Paraguay were collected in the project FAPESP (grant no. 2018/26230-6).

Acknowledgements. John S. Peel (Uppsala University) kindly provided the specimens of *Cambroctoconus koori* for this study. Michael Engelbretsen (Macquarie University) helped T.-Y.S.P and Y.Y.Z. take images of *Tretocylichne perplexa* specimens. Desmond Strusz and Lynne Bean (Australian National University) helped us deal with the specimens housed in Australian National University. Jae Hwa Jin and Junho Kim (KIGAM) allowed us to use the MicroCT equipment housed in KIGAM. Clarissa G. Molinari kindly shared the images of coronate scyphozoans. This is also a contribution to IGCP668: Equatorial Gondwanan History and Early Palaeozoic Evolutionary Dynamics. Constructive and critical suggestions from James Schiffbauer, an anonymous referee, and the associate editor Allison Daley significantly improved the earlier version of the manuscript.

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
