## [Peer Review File · Royal Society Open Science]

Review History

RSOS-210829.R0 (Original submission)

Review form: Reviewer 1

Is the manuscript scientifically sound in its present form?

Yes

Are the interpretations and conclusions justified by the results?

Yes

Is the language acceptable?

Yes

Do you have any ethical concerns with this paper?

No

Have you any concerns about statistical analyses in this paper?

No

Recommendation?

Accept with minor revision (please list in comments)

Comments to the Author(s)

This study incorporates cloudinids, an enigmatic group of tubular macroorganisms widespread during the Ediacaran, into a phylogenetic analysis alongside organisms from the Cambrian interpreted as relatives of the Cnidaria. It thereby constructs a phylogenetic hypothesis in which the cloudinids of the Ediacaran-earliest Cambrian are the early representatives of a clade which persisted into the Miaolingian, attaining larger sizes and more robust skeletons along the way. As this manuscript states, there is considerable uncertainty inherent in reconstructing phylogenetic relationships from such morphologically simple organisms, but taken at face value the evolutionary history outlined here is a unique case study of how an Ediacaran clade adapted to the evolutionary pressures of the Phanerozoic. Viewed more sceptically, it provides a useful, novel hypothesis to guide further research.

Comments:

Page 3, Line 25: 'Putative synapomorphies exist' – Please add a reference or two here, so that it's clear what these synapomorphies are.

Page 3, Lines 41-48: "For example, the branching...triploblastic groups" – I think remove the 'Nevertheless' between these two sentences, as it implies that the second contradicts the first.

Page 3, Line 58: "earliest known usage of skeletal calcium carbonate" – I'd have thought that skeletal microstructure might be a useful feature for building a phylogenetic tree like this, but it's not considered in this study. Could you add a brief explanation of why it was not included?

Page 4, Lines 21-27: "here we present a phylogenetic linkage... were closely related to cloudinids". I think there is a bit missing from this sentence – perhaps it should say "here we present a phylogenetic linkage which shows that cnidarian-related organisms... were closely related to cloudinids". I do think, though, that "closely related" is quite a vague statement of what the results mean, and it might be better to state what the degree of relatedness was.

Page 5, Line 34: I think "Cannon" should be "Canon".

Page 6, Line 7: I would like the manuscript to explain why Sinotubulites was selected as the outgroup in the phylogenetic analyses, and why only one outgroup was used. I think an explanation of outgroup choice would also make the resulting tree easier to interpret. For example, the claim that the cloudinids are 'closely related' to the CLT clade presumably refers to them being more closely related than Sinotubulites is to the CLT clade – but what is the significance of this, if Sinotubulites is the outgroup by definition?

Page 7, Lines 36-37: "The type specimen...Terreneuvian in age [25]" – Please double-check that reference 25 is the appropriate one here.

Page 7, Lines 55-57: "The genus was described as colonial... as solitary individuals" – Please give references, or describe observations in support of this statement.

Page 9, Lines 33-38: "Lipopora lissa...smaller than that of the parental element". Please figure this in Cloudina, or cite a paper that does.

Page 9, Lines 40-43: "This was most likely formed...only known in cnidarians" – Scrutton (1987) describes and figures longitudinal fission occurring in sclerosponges. Is this the same kind of longitudinal fission as is being described here?

Scrutton, C.T., 1987, 'A review of favositid affinities', *Palaeontology* 30, 3, 485-492

Page 9, Line 45: "Cloudinomorphs" should be changed to "Cloudinomorpha".

Page 14, Line 55: "charcteristics" should be "characteristics"

Page 16, Line 34 and 36: "Zunnia" should be changed to "Zuunia"

Page 16, Line 46 – page 17, Line 21: I think the hypothesis being hinted at here is that the non-cloudinid cloudinomorpha are scyphozoans, which is intriguing but needs to be spelled out more clearly.

Page 17, Line 23-26: "the first appearance of external hard parts in Earth's history", this should probably be changed to "the first appearance of external hard parts in macroorganisms", as microfossils with biomineralisation occur before this time.

Page 17, Line 49-51: "making cloudinids the first animals of Ediacaran origin demonstrated to have diversified successfully in the Cambrian". I don't think that this is really the first time this has been demonstrated. If you find any Ediacaran fossils interpreted as animals to be convincing, you have to take the position that they also share a common ancestor with organisms which survived the Ediacaran-Cambrian transition and diversified in the Cambrian too. I think what's interesting about this study is that it gives us more detail on how diversification progressed within one of these groups.

Page 18, Lines 5-41: Are any of these changes statistically significant? Given that these trait changes (increasing size and increasing thickness/aperture) are key aspects of the trait evolution described in this paper, it would be worth backing them up with statistical tests. I think a Mann-Whitney U test would be a suitable way to compare pairs of time intervals and see whether the increases are significant. Also, please give the sample size on each box plot – it is difficult to evaluate them without that information.

Supplementary Material:

In the character matrix, "Sinotabulites" should be "Sinotubulites"

Also, the non-cloudinid cloudinomorphs are included in the supplementary phylogenetic trees, but not in the character matrix. Why is this?

Review form: Reviewer 2 (James Schiffbauer)

Is the manuscript scientifically sound in its present form?

Yes

Are the interpretations and conclusions justified by the results?

No

Is the language acceptable?

No

Do you have any ethical concerns with this paper?

No

Have you any concerns about statistical analyses in this paper?

No

Recommendation?

Major revision is needed (please make suggestions in comments)

Comments to the Author(s)

Review of "Enduring evolutionary embellishment of cloudinids: Lifting the veil of the Cambrian Explosion", Tae-Yoon Park and co-authors, provided by James D. Schiffbauer, University of Missouri

Dear colleagues,

Let me first state up front that, as a general rule, I do not wish to remain anonymous, and I encourage the authors to contact me with any questions they may have regarding my review. I

will also note that this is my 2nd time reviewing this manuscript, albeit for a different journal. While I noticed that some modifications had been incorporated as compared to what I remember from the previous version, I unfortunately cannot say that my main criticism was addressed.

To summarize the contribution, this manuscript strives to phylogenetically link some cloudinomorph taxa (those resembling *Cloudina* in “funnel-in-funnel” construction) from the terminal Ediacaran with the “CLT clade” of cnidarians, *Cambroctoconus*, *Lipopora*, and *Tretocylichne* from Series 2 of the Cambrian Period. The connection between the cloudinomorphs and the CLT clade largely based on presumed similarities in tube construction between these ~30 million-year removed fossils. The authors have compiled an excellent dataset, still almost entirely relegated to the supplement (though I believe Fig. 1 has been expanded herein), showing new photographs, electron micrographs, and some μ CT scans of *Cloudina carinata*, *C. lucianoii*, *Lipopora daseia*, *L. lissa*, *Tretocylichne perplexa*, *Cambroctoconus coreaensis*, and *Ca. koori*, along with morphological assessments and to-scale artistic reconstructions of other published cloudinomorph taxa (which are still excellent, by the way, and I would like to see some of these included in the main text!). I certainly appreciate the correlative microscopic approach, and this unarguably shows a substantial amount of work - here, used by the authors to identify and construct a character matrix for phylogenetic analysis. I don't doubt that this contribution can be quite useful, and I think would help to continue the conversation of the importance of these cloudinomorph organisms in the history of animal life.

My primary concern still exists in this version of the manuscript, however. Namely, that the morphological characters used to connect CLT taxa with cloudinomorphs are undoubtedly convergent and can be found in many terminal Ediacaran and early Cambrian tubular taxa. As I had mentioned before, for example, the exterior tube-wall angularities and polygonal cross-sections/symmetries noted in some cloudinomorphs and the CLT taxa are also observed in *Sinotubulites* (e.g., Cai et al., 2015, *Precambrian Research*) - ranging from triangular to hexagonal shapes. Yet, the authors make only a single mention of *Sinotubulites*, including it as an outgroup in their phylogenetic analysis. In addition, of the organisms studied, only *Cloudina carinata* and *Tretocylichne perplexa* (and maybe *Cloudina lucianoii*) show such tubular angularities. It seems that this morphotype is an oddity of the CLT clade and also of the cloudinomorphs, so I'm left at a bit of a loss for how much weight it could contribute to phylogenetic analysis - yet it seems of utmost importance in the discussion on the continuity of the cloudinids into the Cambrian Series 2.

The authors also note that *Conotubus* and a few other genera (*Saarina*, *Rajatubulus*, *Costatubus*, and *Zuunia*) should be classified as “non-cloudinid cloudinomorphs”, though only some of these were included in the phylogenetic analyses. The authors distinguish from “true cloudinids” which are distinct from the new term they offer, as above, the “non-cloudinid cloudinomorphs”. Though, this distinction itself is problematic. The idea of grouping nested tubular fossils together in a morphological grouping was indeed that phylogenetic relationships are exceedingly difficult to parse, if at all, because of the abundance of tubular forms at this timeframe that have generally comparable construction. Thus, this the introduction of additional terminology that combines form-grouping with phylogeny is not only problematic but also further muddies the water with respect to how these organisms can and should be treated. Without much soft-tissue evidence or further depth of knowledge of tubular ultrastructures to connect various genera, claiming certainty in phylogenetic relations isn't much more than conjecture. The authors claim that “the striking morphological similarity between the CLT clade and cloudinids evince a cnidarian affinity of cloudinids”, but I would argue that we are dealing with known evolutionarily convergent tube-building life modes and morphologies, and thus those similarities instead only evince a successful life mode. For example, we are well aware that tube-dwelling polychaete worms and cnidarians possess these life modes, and the presence of probable through-guts supports the former while polytomous branching the latter. The abundance of funnel- and cup-

like and tubular fossils in the terminal Ediacaran and early Cambrian coupled with the lack of many significant morphological characters or other known soft tissues leaves us, in my opinion, without enough information to know exactly how they may be related.

I also note that some of the Cambrian taxa targeted here, including *Tretocylichne* and *Cambroctoconus*, do not typically show substantial stacking in their morphology – where typically only stacking of a couple units versus tens is common in cloudinomorphs. Rather, *Tretocylichne* and *Cambroctoconus* show smaller cups rim-adjacent to larger basal cups or branching from larger basal cups, which is a construction unlike the broader cloudinomorphs. I will note, however, that this argument is not true for *Lipopora*, but it calls more question onto the so-called “striking morphological similarities”.

As I also mentioned last time through, I feel as though too much information is relegated to the supplement, and the main text doesn't stand enough alone. This leaves me with the suggestion that this manuscript would work better as an integrated but longer manuscript that can incorporate the wealth of data and figures currently stashed in the SOM and targeted for a disciplinary journal, where I feel that it would have more impact on the scientific discourse of the cloudinomorphs.

I will note, quite importantly, that I do like the discussion that this manuscript presents, however. And for that reason, I am suggesting major revision. I think the authors can decide if a disciplinary journal or the forum available at Royal Society Open Science, if they can merge pertinent information from the supplement into the main text, would provide a better platform for this work.

I hope my comments are useful in the preparation of a revision. Should the authors have any questions regarding my review, please feel free to reach out to me - I would welcome the opportunity to discuss this work further. I hope you are all safe and well with the continued pandemic.

All the best,
-Jim

Decision letter (RSOS-210829.R0)

Dear Dr Park

The Editors assigned to your paper RSOS-210829 "ENDURING EVOLUTIONARY EMBELLISHMENT OF CLOUDINIDS IN THE CAMBRIAN" have now received comments from reviewers and would like you to revise the paper in accordance with the reviewer comments and any comments from the Editors. Please note this decision does not guarantee eventual acceptance.

Please submit your revised manuscript and required files (see below) no later than 21 days from today's (ie 02-Sep-2021) date. Note: the ScholarOne system will 'lock' if submission of the revision is attempted 21 or more days after the deadline. If you do not think you will be able to meet this deadline please contact the editorial office immediately.

on behalf of Professor Allison Daley (Associate Editor) and Kevin Padian (Subject Editor)
openscience@royalsociety.org

Associate Editor Comments to Author (Professor Allison Daley):

Associate Editor: 1

Comments to the Author:

The reviewers have both evaluated this manuscript to be an important contribution to the field, although some major revisions are necessary to address their comments. In addition to their recommendations, I provide a more detailed assessment of the phylogenetic methods used. Both reviewers mention that the dataset constructed of morphological observations is excellent and a valuable contribution to the field, and reviewer 2 suggests that including more of these data in the main taxa, rather than the supplement, would clarify the paper and make it a more useful contribution. The discussion is also well formulated and interesting, although Reviewer 1 suggests that more statistical analyses are needed to evaluate the trait evolution through time, and suggests the use of Mann-Whitney U tests to achieve this.

Both reviewers have major concerns about choice of taxa used in the phylogenetic analysis. They suggest that the close relationship of cloudinids and the CLT clade is an artefact of the taxon chosen for inclusion, the choice of the single outgroup, and the taxa that were left out of the analysis but without much justification (Add Jim's taxa here). *Sinotublites* needs to be better justified as the outgroup, and better justification needs to be provided as to why some taxa were disregarded because of convergent tube morphologies, while others were determined to be phylogenetically related. As noted by reviewer 1, too much weight seems to be placed on morphological oddities in the CLT clade and cloudinomorphs (exterior tube-wall angularities and polygonal cross-sections/symmetries), without the authors having included a full

description of the variation in this character more widely in terminal Ediacaran and early Cambrian taxa.

Having evaluated the phylogenetic analysis myself, I wish to note that the methods used in this manuscript don't acknowledge that there is much debate about the most suitable method to analyse discrete morphological characters (O'Reilly et al. 2016; 2018 Palaeontology, including the comment by Goloboff et al. 2018 and their reply; 2018b; Puttick et al. 2017; 2019; Goloboff et al. 2018 Cladistics). Generally, it is recommended that parsimony, maximum likelihood and Bayesian approaches are all applied, compared and discussed. I note that this manuscript only uses only parsimony analysis under equal weights, which is correct but a rather limited approach. Typically, when using parsimony, both equal weights and implied weights should be employed. Only equal weighting was employed in this manuscript, so implied weighting is also recommended (being sure to mention the concavity constants). The software used by the authors, TNT, is the current standard parsimony program used, however more data needs to be provided in methods (e.g. which search function was used; which settings were used – default or otherwise). I also note that the manuscript text mentions 23 characters, however only 22 were described and coded in the Supplementary Information. The phylogenetic character matrix should also be provided in NEXUS format as a supplementary data file. The analysis also appears to lack any assessment of clade support, which is typically provided by Jack-knife sampling (Farris et al. 1996) for example. The authors are also strongly recommended to explore maximum likelihood and/or Bayesian inference methods, for example using IQ-Tree for the former and MrBayes for the latter. If the authors choose to stick with just parsimony, please provide a justification for this methodological choice.

I look forward to seeing the revised version of this manuscript.

Reviewer comments to Author:

Reviewer: 1

Comments to the Author(s)

This study incorporates cloudinids, an enigmatic group of tubular macroorganisms widespread during the Ediacaran, into a phylogenetic analysis alongside organisms from the Cambrian interpreted as relatives of the Cnidaria. It thereby constructs a phylogenetic hypothesis in which the cloudinids of the Ediacaran-earliest Cambrian are the early representatives of a clade which persisted into the Miaolingian, attaining larger sizes and more robust skeletons along the way. As this manuscript states, there is considerable uncertainty inherent in reconstructing phylogenetic relationships from such morphologically simple organisms, but taken at face value the evolutionary history outlined here is a unique case study of how an Ediacaran clade adapted to the evolutionary pressures of the Phanerozoic. Viewed more sceptically, it provides a useful, novel hypothesis to guide further research.

Comments:

Page 3, Line 25: 'Putative synapomorphies exist' – Please add a reference or two here, so that it's clear what these synapomorphies are.

Page 3, Lines 41-48: "For example, the branching...triploblastic groups" – I think remove the 'Nevertheless' between these two sentences, as it implies that the second contradicts the first.

Page 3, Line 58: "earliest known usage of skeletal calcium carbonate" – I'd have thought that skeletal microstructure might be a useful feature for building a phylogenetic tree like this, but it's not considered in this study. Could you add a brief explanation of why it was not included?

Page 4, Lines 21-27: "here we present a phylogenetic linkage... were closely related to cloudinids". I think there is a bit missing from this sentence – perhaps it should say "here we present a phylogenetic linkage which shows that cnidarian-related organisms... were closely

related to cloudinids". I do think, though, that "closely related" is quite a vague statement of what the results mean, and it might be better to state what the degree of relatedness was.

Page 5, Line 34: I think "Cannon" should be "Canon".

Page 6, Line 7: I would like the manuscript to explain why *Sinotubulites* was selected as the outgroup in the phylogenetic analyses, and why only one outgroup was used. I think an explanation of outgroup choice would also make the resulting tree easier to interpret. For example, the claim that the cloudinids are 'closely related' to the CLT clade presumably refers to them being more closely related than *Sinotubulites* is to the CLT clade - but what is the significance of this, if *Sinotubulites* is the outgroup by definition?

Page 7, Lines 36-37: "The type specimen...Terreneuvian in age [25]" - Please double-check that reference 25 is the appropriate one here.

Page 7, Lines 55-57: "The genus was described as colonial... as solitary individuals" - Please give references, or describe observations in support of this statement.

Page 9, Lines 33-38: "*Lipopora lissa*...smaller than that of the parental element". Please figure this in *Cloudina*, or cite a paper that does.

Page 9, Lines 40-43: "This was most likely formed...only known in cnidarians" - Scrutton (1987) describes and figures longitudinal fission occurring in sclerosponges. Is this the same kind of longitudinal fission as is being described here?

Scrutton, C.T., 1987, 'A review of favositid affinities', *Palaeontology* 30, 3, 485-492

Page 9, Line 45: "Cloudinomorphs" should be changed to "Cloudinomorpha".

Page 14, Line 55: "charcteristics" should be "characteristics"

Page 16, Line 34 and 36: "Zunna" should be changed to "Zuunia"

Page 16, Line 46 - page 17, Line 21: I think the hypothesis being hinted at here is that the non-cloudinid cloudinomorpha are scyphozoans, which is intriguing but needs to be spelled out more clearly.

Page 17, Line 23-26: "the first appearance of external hard parts in Earth's history", this should probably be changed to "the first appearance of external hard parts in macroorganisms", as microfossils with biomineralisation occur before this time.

Page 17, Line 49-51: "making cloudinids the first animals of Ediacaran origin demonstrated to have diversified successfully in the Cambrian". I don't think that this is really the first time this has been demonstrated. If you find any Ediacaran fossils interpreted as animals to be convincing, you have to take the position that they also share a common ancestor with organisms which survived the Ediacaran-Cambrian transition and diversified in the Cambrian too. I think what's interesting about this study is that it gives us more detail on how diversification progressed within one of these groups.

Page 18, Lines 5-41: Are any of these changes statistically significant? Given that these trait changes (increasing size and increasing thickness/aperture) are key aspects of the trait evolution described in this paper, it would be worth backing them up with statistical tests. I think a Mann-Whitney U test would be a suitable way to compare pairs of time intervals and see whether the increases are significant. Also, please give the sample size on each box plot - it is difficult to evaluate them without that information.

Supplementary Material:

In the character matrix, "*Sinotubulites*" should be "*Sinotubulites*"

Also, the non-cloudinid cloudinomorpha are included in the supplementary phylogenetic trees, but not in the character matrix. Why is this?

Reviewer: 2

Comments to the Author(s)

Review of "Enduring evolutionary embellishment of cloudinids: Lifting the veil of the Cambrian Explosion", Tae-Yoon Park and co-authors, provided by James D. Schiffbauer, University of Missouri

Dear colleagues,

Let me first state up front that, as a general rule, I do not wish to remain anonymous, and I encourage the authors to contact me with any questions they may have regarding my review. I will also note that this is my 2nd time reviewing this manuscript, albeit for a different journal. While I noticed that some modifications had been incorporated as compared to what I remember from the previous version, I unfortunately cannot say that my main criticism was addressed.

To summarize the contribution, this manuscript strives to phylogenetically link some cloudinomorphic taxa (those resembling *Cloudina* in “funnel-in-funnel” construction) from the terminal Ediacaran with the “CLT clade” of cnidarians, *Cambroctoconus*, *Lipopora*, and *Tretocylichne* from Series 2 of the Cambrian Period. The connection between the cloudinomorphs and the CLT clade largely based on presumed similarities in tube construction between these ~30 million-year removed fossils. The authors have compiled an excellent dataset, still almost entirely relegated to the supplement (though I believe Fig. 1 has been expanded herein), showing new photographs, electron micrographs, and some μ CT scans of *Cloudina carinata*, *C. lucianoii*, *Lipopora daseia*, *L. lissa*, *Tretocylichne perplexa*, *Cambroctoconus coreaensis*, and *Ca. koori*, along with morphological assessments and to-scale artistic reconstructions of other published cloudinomorphic taxa (which are still excellent, by the way, and I would like to see some of these included in the main text!). I certainly appreciate the correlative microscopic approach, and this unarguably shows a substantial amount of work - here, used by the authors to identify and construct a character matrix for phylogenetic analysis. I don't doubt that this contribution can be quite useful, and I think would help to continue the conversation of the importance of these cloudinomorphic organisms in the history of animal life.

My primary concern still exists in this version of the manuscript, however. Namely, that the morphological characters used to connect CLT taxa with cloudinomorphs are undoubtedly convergent and can be found in many terminal Ediacaran and early Cambrian tubular taxa. As I had mentioned before, for example, the exterior tube-wall angularities and polygonal cross-sections/symmetries noted in some cloudinomorphs and the CLT taxa are also observed in *Sinotubulites* (e.g., Cai et al., 2015, *Precambrian Research*) - ranging from triangular to hexagonal shapes. Yet, the authors make only a single mention of *Sinotubulites*, including it as an outgroup in their phylogenetic analysis. In addition, of the organisms studied, only *Cloudina carinata* and *Tretocylichne perplexa* (and maybe *Cloudina lucianoii*) show such tubular angularities. It seems that this morphotype is an oddity of the CLT clade and also of the cloudinomorphs, so I'm left at a bit of a loss for how much weight it could contribute to phylogenetic analysis - yet it seems of utmost importance in the discussion on the continuity of the cloudinids into the Cambrian Series 2.

The authors also note that *Conotubus* and a few other genera (*Saarina*, *Rajatubulus*, *Costatubus*, and *Zuunia*) should be classified as “non-cloudinid cloudinomorphs”, though only some of these were included in the phylogenetic analyses. The authors distinguish from “true cloudinids” which are distinct from the new term they offer, as above, the “non-cloudinid cloudinomorphs”. Though, this distinction itself is problematic. The idea of grouping nested tubular fossils together in a morphological grouping was indeed that phylogenetic relationships are exceedingly difficult to parse, if at all, because of the abundance of tubular forms at this timeframe that have generally comparable construction. Thus, this the introduction of additional terminology that combines form-grouping with phylogeny is not only problematic but also further muddies the water with respect to how these organisms can and should be treated. Without much soft-tissue evidence or further depth of knowledge of tubular ultrastructures to connect various genera, claiming certainty in phylogenetic relations isn't much more than conjecture. The authors claim that “the striking morphological similarity between the CLT clade and cloudinids evince a cnidarian

affinity of cloudinids”, but I would argue that we are dealing with known evolutionarily convergent tube-building life modes and morphologies, and thus those similarities instead only evince a successful life mode. For example, we are well aware that tube-dwelling polychaete worms and cnidarians possess these life modes, and the presence of probable through-guts supports the former while polytomous branching the latter. The abundance of funnel- and cup-like and tubular fossils in the terminal Ediacaran and early Cambrian coupled with the lack of many significant morphological characters or other known soft tissues leaves us, in my opinion, without enough information to know exactly how they may be related.

I also note that some of the Cambrian taxa targeted here, including *Tretocylichne* and *Cambroctoconus*, do not typically show substantial stacking in their morphology – where typically only stacking of a couple units versus tens is common in cloudinomorphs. Rather, *Tretocylichne* and *Cambroctoconus* show smaller cups rim-adjacent to larger basal cups or branching from larger basal cups, which is a construction unlike the broader cloudinomorphs. I will note, however, that this argument is not true for *Lipopora*, but it calls more question onto the so-called “striking morphological similarities”.

As I also mentioned last time through, I feel as though too much information is relegated to the supplement, and the main text doesn’t stand enough alone. This leaves me with the suggestion that this manuscript would work better as an integrated but longer manuscript that can incorporate the wealth of data and figures currently stashed in the SOM and targeted for a disciplinary journal, where I feel that it would have more impact on the scientific discourse of the cloudinomorphs.

I will note, quite importantly, that I do like the discussion that this manuscript presents, however. And for that reason, I am suggesting major revision. I think the authors can decide if a disciplinary journal or the forum available at Royal Society Open Science, if they can merge pertinent information from the supplement into the main text, would provide a better platform for this work.

I hope my comments are useful in the preparation of a revision. Should the authors have any questions regarding my review, please feel free to reach out to me - I would welcome the opportunity to discuss this work further. I hope you are all safe and well with the continued pandemic.

All the best,
-Jim

===PREPARING YOUR MANUSCRIPT===

Your revised paper should include the changes requested by the referees and Editors of your manuscript. You should provide two versions of this manuscript and both versions must be provided in an editable format:
one version identifying all the changes that have been made (for instance, in coloured highlight, in bold text, or tracked changes);
a 'clean' version of the new manuscript that incorporates the changes made, but does not highlight them. This version will be used for typesetting if your manuscript is accepted.
Please ensure that any equations included in the paper are editable text and not embedded images.

Please ensure that you include an acknowledgements' section before your reference list/bibliography. This should acknowledge anyone who assisted with your work, but does not

qualify as an author per the guidelines at <https://royalsociety.org/journals/ethics-policies/openness/>.

===PREPARING YOUR REVISION IN SCHOLARONE===

- Ensure that your data access statement meets the requirements at <https://royalsociety.org/journals/authors/author-guidelines/#data>. You should ensure that you cite the dataset in your reference list. If you have deposited data etc in the Dryad repository, please include both the 'For publication' link and 'For review' link at this stage.
- If you are requesting an article processing charge waiver, you must select the relevant waiver option (if requesting a discretionary waiver, the form should have been uploaded at Step 3 'File upload' above).
- If you have uploaded ESM files, please ensure you follow the guidance at <https://royalsociety.org/journals/authors/author-guidelines/#supplementary-material> to include a suitable title and informative caption. An example of appropriate titling and captioning may be found at https://figshare.com/articles/Table_S2_from_Is_there_a_trade-off_between_peak_performance_and_performance_breadth_across_temperatures_for_aerobic_scooping_in_teleost_fishes_/3843624.

Author's Response to Decision Letter for (RSOS-210829.R0)

See Appendix A.

Decision letter (RSOS-210829.R1)

Dear Dr Park

On behalf of the Editors, we are pleased to inform you that your Manuscript RSOS-210829.R1 "ENDURING EVOLUTIONARY EMBELLISHMENT OF CLOUDINIDS IN THE CAMBRIAN" has been accepted for publication in Royal Society Open Science subject to minor revision in accordance with the referees' reports. Please find the referees' comments along with any feedback from the Editors below my signature.

Please submit your revised manuscript and required files (see below) no later than 7 days from today's (ie 08-Nov-2021) date. Note: the ScholarOne system will 'lock' if submission of the revision is attempted 7 or more days after the deadline. If you do not think you will be able to meet this deadline please contact the editorial office immediately.

on behalf of Professor Allison Daley (Associate Editor) and Kevin Padian (Subject Editor)
openscience@royalsociety.org

Associate Editor Comments to Author (Professor Allison Daley):
Thank you for your detailed revisions to this manuscript. All major comments have been addressed well, and I recommend acceptance after the following minor addition. In your acknowledgements section, please extend your thanks to the reviewers (anonymous and named) and anyone else who provided comments on your manuscript. From the reviewers comments and your own, it seems the manuscript has evolved during numerous rounds of reviews and revisions, and the constructive comments from reviewers should be acknowledged.

===PREPARING YOUR MANUSCRIPT===

You should provide two versions of this manuscript and both versions must be provided in an editable format:
one version should clearly identify all the changes that have been made (for instance, in coloured highlight, in bold text, or tracked changes);
a 'clean' version of the new manuscript that incorporates the changes made, but does not highlight them. This version will be used for typesetting.

===PREPARING YOUR REVISION IN SCHOLARONE===

<https://royalsociety.org/journals/authors/author-guidelines/#data>. You should ensure that

you cite the dataset in your reference list. If you have deposited data etc in the Dryad repository, please only include the 'For publication' link at this stage. You should remove the 'For review' link.

-- If you are requesting an article processing charge waiver, you must select the relevant waiver option (if requesting a discretionary waiver, the form should have been uploaded, see 'File upload' above).

-- If you have uploaded any electronic supplementary (ESM) files, please ensure you follow the guidance at <https://royalsociety.org/journals/authors/author-guidelines/#supplementary-material> to include a suitable title and informative caption. An example of appropriate titling and captioning may be found at https://figshare.com/articles/Table_S2_from_Is_there_a_trade-off_between_peak_performance_and_performance_breadth_across_temperatures_for_aerobic_scope_in_teleost_fishes_/3843624.

Author's Response to Decision Letter for (RSOS-210829.R1)

See Appendix B.

Decision letter (RSOS-210829.R2)

Dear Dr Park,

I am pleased to inform you that your manuscript entitled "ENDURING EVOLUTIONARY EMBELLISHMENT OF CLOUDINIDS IN THE CAMBRIAN" is now accepted for publication in Royal Society Open Science.

The proof of your paper will be available for review using the Royal Society online proofing system and you will receive details of how to access this in the near future from our production office (openscience_proofs@royalsociety.org). We aim to maintain rapid times to publication after

acceptance of your manuscript and we would ask you to please contact both the production office and editorial office if you are likely to be away from e-mail contact to minimise delays to publication. If you are going to be away, please nominate a co-author (if available) to manage the proofing process, and ensure they are copied into your email to the journal.

on behalf of Professor Allison Daley (Associate Editor) and Kevin Padian (Subject Editor)
openscience@royalsociety.org

Appendix A

Dear Editor

This is a note on our responses to the very helpful comments by the reviewers and you as editor on our manuscript: **Enduring evolutionary embellishment of cloudinids in the Cambrian.**

First of all, we greatly appreciate your and the reviewers' constructive and positive comments. We have followed all suggestions as comprehensively as possible and indicate our actions below, including our justification for the queries regarding some of our interpretations. Reviewers' comments are written in black, while our replies are marked in blue.

Associate Editor's comments

Having evaluated the phylogenetic analysis myself, I wish to note that the methods used in this manuscript don't acknowledge that there is much debate about the most suitable method to analyse discrete morphological characters (O'Reilly et al. 2016; 2018 *Palaeontology*, including the comment by Goloboff et al. 2018 and their reply; 2018b; Puttick et al. 2017; 2019; Goloboff et al. 2018 *Cladistics*). Generally, it is recommended that parsimony, maximum likelihood and Bayesian approaches are all applied, compared and discussed. I note that this manuscript only uses only parsimony analysis under equal weights, which is correct but a rather limited approach. Typically, when using parsimony, both equal weights and implied weights should be employed. Only equal weighting was employed in this manuscript, so implied weighting is also recommended (being sure to mention the concavity constants). The software used by the authors, TNT, is the current standard parsimony program used, however more data needs to be provided in methods (e.g. which search function was used; which settings were used – default or otherwise). I also note that the manuscript text mentions 23 characters, however only 22 were described and coded in the Supplementary Information.

The phylogenetic character matrix should also be provided in NEXUS format as a supplementary data file. The analysis also appears to lack any assessment of clade support, which is typically provided by Jack-knife sampling (Farris et al. 1996) for example. The authors are also strongly recommended to explore maximum likelihood and/or Bayesian inference methods, for example using IQ-Tree for the former and MrBayes for the latter. If the authors choose to stick with just parsimony, please provide a justification for this methodological choice.

Thank you for this sensible suggestion. We have now run the Maximum Likelihood and Bayesian analyses requested, and included the results as an additional supplementary figure (S5). For parsimony analysis, given the limited number of characters, we think equal weighting is appropriate. The results of these different methods all concord with the arguments we are advancing in this paper.

We have added more details of each analysis, including the search function and settings in the methods section, and fixed the discrepancy in number of characters mentioned in text and data matrix of Supplementary Information. We also have uploaded the phylogenetic character matrix (NEXUS file) to the Dryad, which are recommended data repository website by editorial board. Thank you very much for noticing this.

Reviewer1's comments:

Comments:

Page 3, Line 25: 'Putative synapomorphies exist' – Please add a reference or two here, so that it's clear what these synapomorphies are.

We have added a reference (Hoyal Cuthill and Han 2018) for a reference.

Page 3, Lines 41-48: "For example, the branching...triploblastic groups" – I think remove the 'Nevertheless' between these two sentences, as it implies that the second contradicts the first.

Emended as suggested.

Page 3, Line 58: "earliest known usage of skeletal calcium carbonate" - I'd have thought that skeletal microstructure might be a useful feature for building a phylogenetic tree like this, but it's not considered in this study. Could you add a brief explanation of why it was not included?

This is a promising possibility when more material becomes available. However, it was practically impossible for us to delve into the microstructure of all the species included in the study. For example, the only *Lipopora* and *Tritocylichne* specimens available to us were type specimens, so that we were not allowed to work further for their microstructure.

Page 4, Lines 21-27: "here we present a phylogenetic linkage... were closely related to cloudinids". I think there is a bit missing from this sentence – perhaps it should say "here we present a phylogenetic linkage which shows that cnidarian-related organisms... were closely related to cloudinids". I do think, though, that "closely related" is quite a vague statement of what the results mean, and it might be better to state what the degree of relatedness was.

Thank you for noticing this. The text has been emended as suggested. As for 'closely-related', we think this is the least controversial way of saying it. But as suggested, we have added 'probably descended from' for more clarity.

Page 5, Line 34: I think "Cannon" should be "Canon".

Emended as suggested.

Page 6, Line 7: I would like the manuscript to explain why *Sinotubulites* was selected as the outgroup in the phylogenetic analyses, and why only one outgroup was used. I think an explanation of outgroup choice would also make the resulting tree easier to interpret. For example, the claim that the cloudinids are ‘closely related’ to the CLT clade presumably refers to them being more closely related than *Sinotubulites* is to the CLT clade – but what is the significance of this, if *Sinotubulites* is the outgroup by definition?

Thank you for the suggestion. We have added a sentence explaining why *Sinotubulites* was selected, which now states “*Sinotubulites* was selected as an outgroup, since it is a reasonably well known tubular fossil occurring in the latest Ediacaran, but lacked the typical funnel-in-funnel structure of cloudinids.” We were reluctant to use multiple outgroups, since there has not been any phylogenetic analysis of the cloudinids and other Ediacaran tubular organisms, so that we do not have any reliable knowledge to use other outgroup than *Sinotubulites*.

Page 7, Lines 36-37: “The type specimen...Terreneuvian in age [25]” – Please double-check that reference 25 is the appropriate one here.

We have double-checked it. The related story with Alvaro et al. (2020) is also present in the paragraph for *Cloudina carinata*.

Page 7, Lines 55-57: “The genus was described as colonial... as solitary individuals” – Please give references, or describe observations in support of this statement.

We have added Jell and Jell (1976) for the reference.

Page 9, Lines 33-38: “*Lipopora lissa*...smaller than that of the parental element”. Please figure this in *Cloudina*, or cite a paper that does.

We have added Hua et al. (2005) for the reference.

Page 9, Lines 40-43: “This was most likely formed...only known in cnidarians” – Scrutton (1987) describes and figures longitudinal fission occurring in sclerosponges. Is this the same kind of longitudinal fission as is being described here?

Scrutton, C.T., 1987, ‘A review of favositid affinities’, *Palaeontology* 30, 3, 485-492

Thank you for informing this interesting phenomenon. But we think the longitudinal fission of sclerosponge is something distinguished from the longitudinal fission of diploblastic animals.

Page 9, Line 45: “Cloudinomorphs” should be changed to “Cloudinomorphs”.

Emended as suggested. Thank you for noticing.

Page 14, Line 55: “charcteristics” should be “characteristics”

Emended as suggested. Thank you for noticing.

Page 16, Line 34 and 36: “Zunnia” should be changed to “Zuunia”

Emended as suggested. Thank you for noticing.

Page 16, Line 46 – page 17, Line 21: I think the hypothesis being hinted at here is that the non-cloudinid cloudinomorphs are scyphozoans, which is intriguing but needs to be spelled out more clearly.

Thank you for this suggestion. Although this is what we think, our experience with this study has been that such side comments commonly distract readers from the main points we are making. As this is not part of the main thrust of the paper, we prefer to remain cautious.

Page 17, Line 23-26: “the first appearance of external hard parts in Earth’s history”, this should probably be changed to “the first appearance of external hard parts in macroorganisms”, as microfossils with biomineralisation occur before this time.

Emended as suggested. Thank you for the suggestion.

Page 17, Line 49-51: “making cloudinids the first animals of Ediacaran origin demonstrated to have diversified successfully in the Cambrian”. I don’t think that this is really the first time this has been demonstrated. If you find any Ediacaran fossils interpreted as animals to be convincing, you have to take the position that they also share a common ancestor with organisms which survived the Ediacaran-Cambrian transition and diversified in the Cambrian too. I think what’s interesting about this study is that it gives us more detail on how diversification progressed within one of these groups.

Thank you for the suggestion. We agree with it, and emended as ‘making cloudinids the first individual animal clade of Ediacaran origin shown to have diversified successfully in the Cambrian’.

Page 18, Lines 5-41: Are any of these changes statistically significant? Given that these trait changes (increasing size and increasing thickness/aperture) are key aspects of the trait evolution described in this paper, it would be worth backing them up with statistical tests. I think a Mann-Whitney U test would be a suitable way to compare pairs of time intervals and see whether the increases are significant. Also, please give the sample size on each box plot – it is difficult to evaluate them without that information.

We have conducted Mann-Whitney U test, reported it in the main text, and added further details in the Supplementary materials. Also, we have added information on the sample size in the Materials and Methods section.

Supplementary Material:

In the character matrix, “Sinotabulites” should be “Sinotubulites”

Emended as suggested. Thank you for noticing.

Also, the non-cloudinid cloudinomorphs are included in the supplementary phylogenetic

trees, but not in the character matrix. Why is this?

This was our mistake. Thank you for noticing this. We have added data of the five ‘cloudinid-like tubular organisms.’

Reviewer2’ comments

My primary concern still exists in this version of the manuscript, however. Namely, that the morphological characters used to connect CLT taxa with cloudinomorphs are undoubtedly convergent and can be found in many terminal Ediacaran and early Cambrian tubular taxa. As I had mentioned before, for example, the exterior tube-wall angularities and polygonal cross-sections/symmetries noted in some cloudinomorphs and the CLT taxa are also observed in *Sinotubulites* (e.g., Cai et al., 2015, Precambrian Research) - ranging from triangular to hexagonal shapes. Yet, the authors make only a single mention of *Sinotubulites*, including it as an outgroup in their phylogenetic analysis. In addition, of the organisms studied, only *Cloudina carinata* and *Tretocylichne perplexa* (and maybe *Cloudina lucianoii*) show such tubular angularities. It seems that this morphotype is an oddity of the CLT clade and also of the cloudinomorphs, so I'm left at a bit of a loss for how much weight it could contribute to phylogenetic analysis - yet it seems of utmost importance in the discussion on the continuity of the cloudinids into the Cambrian Series 2.

We fully understand this concern. First, as mentioned in response to reviewer 1, we think *Sinotubulites* is similar to cloudinomorphs in having a tubular shape, but is distinguished in lacking the typical funnel-in-funnel structure. The angularity of *Sinotubulites*, however, is rather variable (as mentioned, from triangular to hexagonal), whereas those of *Cloudina carinata* and the CLT clade converge toward octagonal. For us, yes, the octaradial symmetry is a key character of phylogenetic importance, and a plesiomorphic feature of cnidarians. This is a hypothesis of relationships, but it is one that is consistent with the results of our analyses. It is a contribution to and perspective on cloudinid evolution that should be stimulating to the scientific community.

The authors also note that *Conotubus* and a few other genera (*Saarina*, *Rajatubulus*, *Costatubus*, and *Zuunia*) should be classified as “non-cloudinid cloudinomorphs”, though only some of these were included in the phylogenetic analyses. The authors distinguish from “true cloudinids” which are distinct from the new term they offer, as above, the “non-cloudinid cloudinomorphs”. Though, this distinction itself is problematic. The idea of grouping nested tubular fossils together in a morphological grouping was indeed that phylogenetic relationships are exceedingly difficult to parse, if at all, because of the abundance of tubular forms at this timeframe that have generally comparable construction. Thus, this the introduction of additional terminology that combines form-grouping with phylogeny is not only problematic but also further muddies the water with respect to how these organisms can and should be treated. Without much soft-tissue evidence or further depth of knowledge of tubular ultrastructures to connect various genera, claiming certainty in phylogenetic relations isn’t much more than conjecture. The authors claim that “the striking morphological similarity between the CLT clade and cloudinids evince a cnidarian affinity of cloudinids”, but I would argue that we are dealing with known evolutionarily convergent

tube-building life modes and morphologies, and thus those similarities instead only evince a successful life mode. For example, we are well aware that tube-dwelling polychaete worms and cnidarians possess these life modes, and the presence of probable through-guts supports the former while polytomous branching the latter. The abundance of funnel- and cup-like and tubular fossils in the terminal Ediacaran and early Cambrian coupled with the lack of many significant morphological characters or other known soft tissues leaves us, in my opinion, without enough information to know exactly how they may be related.

As reviewer 2 may know, we did not include the groups referred to in this comment in the phylogenetic analysis in previous versions of this manuscripts submitted elsewhere. But along the journey of this MS, we were asked to include them in the analysis in order that they can be distinguished from “true” cloudinids. The main reason why we call them “non-cloudinid cloudinomorphs” is that they have been called “cloudinomorphs” in many publications, and we have not found convincing evidence establishing their relationships with “true” cloudinids (i.e. those considered herein that include the type *Cloudina*). Moreover, we thought that excluding “non-cloudinid cloudinomorphs” from our analysis would be a way to avoid unnecessary debate (or conflict) resulting from recently published papers (*Nature Communications* and *Scientific Reports*) suggesting various bilaterian affinities for those animals. Respecting the reviewer’s opinion, however, we use ‘the cloudinid-like tubular organisms’ to replace the term ‘non-cloudinid cloudinomorphs.’

I also note that some of the Cambrian taxa targeted here, including *Tretocylichne* and *Cambroctoconus*, do not typically show substantial stacking in their morphology – where typically only stacking of a couple units versus tens in common in cloudinomorphs. Rather, *Tretocylichne* and *Cambroctoconus* show smaller cups rim-adjacent to larger basal cups or branching from larger basal cups, which is a construction unlike the broader cloudinomorphs. I will note, however, that this argument is not true for *Lipopora*, but it calls more question onto the so-called “striking morphological similarities”.

When we discovered the funnel-like construction of *Cambroctoconus* which, otherwise, only shows a mode of budding for asexual reproduction, we thought it could possibly have some implication for the origin of *Cambroctoconus*. Then we linked them to the construction mode of *Lipopora lissa* (and *C. carinata*). And it seemed pretty much obvious that *Cambroctoconus*, *Lipopora*, and *Tretocylichne* are closely related to each other based on these similarities that we would say are self-evidently striking. Going through the peer-review processes, we understand that not all may share this view, particularly if they have advocated for alternative affinities. All we are doing here is presenting character-based evidence for a cloudinid affinity with some of least derived metazoan groups known to have flourished in the Phanerozoic.

As I also mentioned last time through, I feel as though too much information is relegated to the supplement, and the main text doesn’t stand enough alone. This leaves me with the suggestion that this manuscript would work better as an integrated but longer manuscript that can incorporate the wealth of data and figures currently stashed in the SOM and targeted for a disciplinary journal, where I feel that it would have more impact on the scientific discourse of the cloudinomorphs.

I will note, quite importantly, that I do like the discussion that this manuscript presents,

however. And for that reason, I am suggesting major revision. I think the authors can decide if a disciplinary journal or the forum available at Royal Society Open Science, if they can merge pertinent information from the supplement into the main text, would provide a better platform for this work.

Compared to the original manuscript that the reviewer2 previously reviewed, much of the supplementary information have been moved into the main figure and main text. For example, best images of each cloudinids have been incorporated into Figure 1, so that the readers could get the essence of morphology of each species at a glance. We think there are currently too many supplementary figures to be incorporated into the main text; now even there are two additional supplementary figures to meet the recommendations during this review process. The detailed descriptions of each “true” cloudinid species have been already incorporated in the main text, leaving only the descriptions for the cloudinids not-considered in this study, and the “non-cloudinid cloudinomorphs (or the cloudinid-like tubular organisms as newly emended).” If incorporating these parts into the main text is still considered as necessary, we are happy to do that in the next round of review, but our feeling is that the paper can have most impact if it is relatively short and to-the-point, allowing readers to access the Supplementary materials if they are stimulated to do so.

Appendix B

Dear Editor

This is a note on our responses to the very helpful comments by the reviewers and you as editor on our manuscript: **Enduring evolutionary embellishment of cloudinids in the Cambrian.**

First of all, we greatly appreciate your and the reviewers' constructive and positive comments. We have followed all suggestions as comprehensively as possible and indicate our actions below, including our justification for the queries regarding some of our interpretations. Reviewers' comments are written in black, while our replies are marked in blue.

Associate Editor's comments

Having evaluated the phylogenetic analysis myself, I wish to note that the methods used in this manuscript don't acknowledge that there is much debate about the most suitable method to analyse discrete morphological characters (O'Reilly et al. 2016; 2018 Palaeontology, including the comment by Goloboff et al. 2018 and their reply; 2018b; Puttick et al. 2017; 2019; Goloboff et al. 2018 Cladistics). Generally, it is recommended that parsimony, maximum likelihood and Bayesian approaches are all applied, compared and discussed. I note that this manuscript only uses only parsimony analysis under equal weights, which is correct but a rather limited approach. Typically, when using parsimony, both equal weights and implied weights should be employed. Only equal weighting was employed in this manuscript, so implied weighting is also recommended (being sure to mention the concavity constants). The software used by the authors, TNT, is the current standard parsimony program used, however more data needs to be provided in methods (e.g. which search function was used; which settings were used – default or otherwise). I also note that the manuscript text mentions 23 characters, however only 22 were described and coded in the Supplementary Information.

The phylogenetic character matrix should also be provided in NEXUS format as a supplementary data file. The analysis also appears to lack any assessment of clade support, which is typically provided by Jack-knife sampling (Farris et al. 1996) for example. The authors are also strongly recommended to explore maximum likelihood and/or Bayesian inference methods, for example using IQ-Tree for the former and MrBayes for the latter. If the authors choose to stick with just parsimony, please provide a justification for this methodological choice.

Thank you for this sensible suggestion. We have now run the Maximum Likelihood and Bayesian analyses requested, and included the results as an additional supplementary figure (S5). For parsimony analysis, given the limited number of characters, we think equal weighting is appropriate. The results of these different methods all concord with the arguments we are advancing in this paper.

We have added more details of each analysis, including the search function and settings in the methods section, and fixed the discrepancy in number of characters mentioned in text and data matrix of Supplementary Information. We also have uploaded the phylogenetic character matrix (NEXUS file) to the Dryad, which are recommended data repository website by editorial board. Thank you very much for noticing this.

Reviewer1's comments:

Comments:

Page 3, Line 25: 'Putative synapomorphies exist' – Please add a reference or two here, so that it's clear what these synapomorphies are.

We have added a reference (Hoyal Cuthill and Han 2018) for a reference.

Page 3, Lines 41-48: "For example, the branching...triploblastic groups" – I think remove the 'Nevertheless' between these two sentences, as it implies that the second contradicts the first.

Emended as suggested.

Page 3, Line 58: "earliest known usage of skeletal calcium carbonate" - I'd have thought that skeletal microstructure might be a useful feature for building a phylogenetic tree like this, but it's not considered in this study. Could you add a brief explanation of why it was not included?

This is a promising possibility when more material becomes available. However, it was practically impossible for us to delve into the microstructure of all the species included in the study. For example, the only *Lipopora* and *Tritocylichne* specimens available to us were type specimens, so that we were not allowed to work further for their microstructure.

Page 4, Lines 21-27: "here we present a phylogenetic linkage... were closely related to cloudinids". I think there is a bit missing from this sentence – perhaps it should say "here we present a phylogenetic linkage which shows that cnidarian-related organisms... were closely related to cloudinids". I do think, though, that "closely related" is quite a vague statement of what the results mean, and it might be better to state what the degree of relatedness was.

Thank you for noticing this. The text has been emended as suggested. As for 'closely-related', we think this is the least controversial way of saying it. But as suggested, we have added 'probably descended from' for more clarity.

Page 5, Line 34: I think "Cannon" should be "Canon".

Emended as suggested.

Page 6, Line 7: I would like the manuscript to explain why *Sinotubulites* was selected as the outgroup in the phylogenetic analyses, and why only one outgroup was used. I think an explanation of outgroup choice would also make the resulting tree easier to interpret. For example, the claim that the cloudinids are ‘closely related’ to the CLT clade presumably refers to them being more closely related than *Sinotubulites* is to the CLT clade – but what is the significance of this, if *Sinotubulites* is the outgroup by definition?

Thank you for the suggestion. We have added a sentence explaining why *Sinotubulites* was selected, which now states “*Sinotubulites* was selected as an outgroup, since it is a reasonably well known tubular fossil occurring in the latest Ediacaran, but lacked the typical funnel-in-funnel structure of cloudinids.” We were reluctant to use multiple outgroups, since there has not been any phylogenetic analysis of the cloudinids and other Ediacaran tubular organisms, so that we do not have any reliable knowledge to use other outgroup than *Sinotubulites*.

Page 7, Lines 36-37: “The type specimen...Terreneuvian in age [25]” – Please double-check that reference 25 is the appropriate one here.

We have double-checked it. The related story with Alvaro et al. (2020) is also present in the paragraph for *Cloudina carinata*.

Page 7, Lines 55-57: “The genus was described as colonial... as solitary individuals” – Please give references, or describe observations in support of this statement.

We have added Jell and Jell (1976) for the reference.

Page 9, Lines 33-38: “*Lipopora lissa*...smaller than that of the parental element”. Please figure this in *Cloudina*, or cite a paper that does.

We have added Hua et al. (2005) for the reference.

Page 9, Lines 40-43: “This was most likely formed...only known in cnidarians” – Scrutton (1987) describes and figures longitudinal fission occurring in sclerosponges. Is this the same kind of longitudinal fission as is being described here?

Scrutton, C.T., 1987, ‘A review of favositid affinities’, *Palaeontology* 30, 3, 485-492

Thank you for informing this interesting phenomenon. But we think the longitudinal fission of sclerosponge is something distinguished from the longitudinal fission of diploblastic animals.

Page 9, Line 45: “Cloudinomorphs” should be changed to “Cloudinomorphs”.

Emended as suggested. Thank you for noticing.

Page 14, Line 55: “charcteristics” should be “characteristics”

Emended as suggested. Thank you for noticing.

Page 16, Line 34 and 36: “Zunnia” should be changed to “Zuunia”

Emended as suggested. Thank you for noticing.

Page 16, Line 46 – page 17, Line 21: I think the hypothesis being hinted at here is that the non-cloudinid cloudinomorphs are scyphozoans, which is intriguing but needs to be spelled out more clearly.

Thank you for this suggestion. Although this is what we think, our experience with this study has been that such side comments commonly distract readers from the main points we are making. As this is not part of the main thrust of the paper, we prefer to remain cautious.

Page 17, Line 23-26: “the first appearance of external hard parts in Earth’s history”, this should probably be changed to “the first appearance of external hard parts in macroorganisms”, as microfossils with biomineralisation occur before this time.

Emended as suggested. Thank you for the suggestion.

Page 17, Line 49-51: “making cloudinids the first animals of Ediacaran origin demonstrated to have diversified successfully in the Cambrian”. I don’t think that this is really the first time this has been demonstrated. If you find any Ediacaran fossils interpreted as animals to be convincing, you have to take the position that they also share a common ancestor with organisms which survived the Ediacaran-Cambrian transition and diversified in the Cambrian too. I think what’s interesting about this study is that it gives us more detail on how diversification progressed within one of these groups.

Thank you for the suggestion. We agree with it, and emended as ‘making cloudinids the first individual animal clade of Ediacaran origin shown to have diversified successfully in the Cambrian’.

Page 18, Lines 5-41: Are any of these changes statistically significant? Given that these trait changes (increasing size and increasing thickness/aperture) are key aspects of the trait evolution described in this paper, it would be worth backing them up with statistical tests. I think a Mann-Whitney U test would be a suitable way to compare pairs of time intervals and see whether the increases are significant. Also, please give the sample size on each box plot – it is difficult to evaluate them without that information.

We have conducted Mann-Whitney U test, reported it in the main text, and added further details in the Supplementary materials. Also, we have added information on the sample size in the Materials and Methods section.

Supplementary Material:

In the character matrix, “Sinotabulites” should be “Sinotubulites”

Emended as suggested. Thank you for noticing.

Also, the non-cloudinid cloudinomorphs are included in the supplementary phylogenetic

trees, but not in the character matrix. Why is this?

This was our mistake. Thank you for noticing this. We have added data of the five ‘cloudinid-like tubular organisms.’

Reviewer2’ comments

My primary concern still exists in this version of the manuscript, however. Namely, that the morphological characters used to connect CLT taxa with cloudinomorphs are undoubtedly convergent and can be found in many terminal Ediacaran and early Cambrian tubular taxa. As I had mentioned before, for example, the exterior tube-wall angularities and polygonal cross-sections/symmetries noted in some cloudinomorphs and the CLT taxa are also observed in *Sinotubulites* (e.g., Cai et al., 2015, Precambrian Research) - ranging from triangular to hexagonal shapes. Yet, the authors make only a single mention of *Sinotubulites*, including it as an outgroup in their phylogenetic analysis. In addition, of the organisms studied, only *Cloudina carinata* and *Tretocylichne perplexa* (and maybe *Cloudina lucianoii*) show such tubular angularities. It seems that this morphotype is an oddity of the CLT clade and also of the cloudinomorphs, so I'm left at a bit of a loss for how much weight it could contribute to phylogenetic analysis - yet it seems of utmost importance in the discussion on the continuity of the cloudinids into the Cambrian Series 2.

We fully understand this concern. First, as mentioned in response to reviewer 1, we think *Sinotubulites* is similar to cloudinomorphs in having a tubular shape, but is distinguished in lacking the typical funnel-in-funnel structure. The angularity of *Sinotubulites*, however, is rather variable (as mentioned, from triangular to hexagonal), whereas those of *Cloudina carinata* and the CLT clade converge toward octagonal. For us, yes, the octaradial symmetry is a key character of phylogenetic importance, and a plesiomorphic feature of cnidarians. This is a hypothesis of relationships, but it is one that is consistent with the results of our analyses. It is a contribution to and perspective on cloudinid evolution that should be stimulating to the scientific community.

The authors also note that *Conotubus* and a few other genera (*Saarina*, *Rajatubulus*, *Costatubus*, and *Zuunia*) should be classified as “non-cloudinid cloudinomorphs”, though only some of these were included in the phylogenetic analyses. The authors distinguish from “true cloudinids” which are distinct from the new term they offer, as above, the “non-cloudinid cloudinomorphs”. Though, this distinction itself is problematic. The idea of grouping nested tubular fossils together in a morphological grouping was indeed that phylogenetic relationships are exceedingly difficult to parse, if at all, because of the abundance of tubular forms at this timeframe that have generally comparable construction. Thus, this the introduction of additional terminology that combines form-grouping with phylogeny is not only problematic but also further muddies the water with respect to how these organisms can and should be treated. Without much soft-tissue evidence or further depth of knowledge of tubular ultrastructures to connect various genera, claiming certainty in phylogenetic relations isn't much more than conjecture. The authors claim that “the striking morphological similarity between the CLT clade and cloudinids evince a cnidarian affinity of cloudinids”, but I would argue that we are dealing with known evolutionarily convergent

tube-building life modes and morphologies, and thus those similarities instead only evince a successful life mode. For example, we are well aware that tube-dwelling polychaete worms and cnidarians possess these life modes, and the presence of probable through-guts supports the former while polytomous branching the latter. The abundance of funnel- and cup-like and tubular fossils in the terminal Ediacaran and early Cambrian coupled with the lack of many significant morphological characters or other known soft tissues leaves us, in my opinion, without enough information to know exactly how they may be related.

As reviewer 2 may know, we did not include the groups referred to in this comment in the phylogenetic analysis in previous versions of this manuscripts submitted elsewhere. But along the journey of this MS, we were asked to include them in the analysis in order that they can be distinguished from “true” cloudinids. The main reason why we call them “non-cloudinid cloudinomorphs” is that they have been called “cloudinomorphs” in many publications, and we have not found convincing evidence establishing their relationships with “true” cloudinids (i.e. those considered herein that include the type *Cloudina*). Moreover, we thought that excluding “non-cloudinid cloudinomorphs” from our analysis would be a way to avoid unnecessary debate (or conflict) resulting from recently published papers (*Nature Communications* and *Scientific Reports*) suggesting various bilaterian affinities for those animals. Respecting the reviewer’s opinion, however, we use ‘the cloudinid-like tubular organisms’ to replace the term ‘non-cloudinid cloudinomorphs.’

I also note that some of the Cambrian taxa targeted here, including *Tretocylichne* and *Cambroctoconus*, do not typically show substantial stacking in their morphology – where typically only stacking of a couple units versus tens in common in cloudinomorphs. Rather, *Tretocylichne* and *Cambroctoconus* show smaller cups rim-adjacent to larger basal cups or branching from larger basal cups, which is a construction unlike the broader cloudinomorphs. I will note, however, that this argument is not true for *Lipopora*, but it calls more question onto the so-called “striking morphological similarities”.

When we discovered the funnel-like construction of *Cambroctoconus* which, otherwise, only shows a mode of budding for asexual reproduction, we thought it could possibly have some implication for the origin of *Cambroctoconus*. Then we linked them to the construction mode of *Lipopora lissa* (and *C. carinata*). And it seemed pretty much obvious that *Cambroctoconus*, *Lipopora*, and *Tretocylichne* are closely related to each other based on these similarities that we would say are self-evidently striking. Going through the peer-review processes, we understand that not all may share this view, particularly if they have advocated for alternative affinities. All we are doing here is presenting character-based evidence for a cloudinid affinity with some of least derived metazoan groups known to have flourished in the Phanerozoic.

As I also mentioned last time through, I feel as though too much information is relegated to the supplement, and the main text doesn’t stand enough alone. This leaves me with the suggestion that this manuscript would work better as an integrated but longer manuscript that can incorporate the wealth of data and figures currently stashed in the SOM and targeted for a disciplinary journal, where I feel that it would have more impact on the scientific discourse of the cloudinomorphs.

I will note, quite importantly, that I do like the discussion that this manuscript presents,

however. And for that reason, I am suggesting major revision. I think the authors can decide if a disciplinary journal or the forum available at Royal Society Open Science, if they can merge pertinent information from the supplement into the main text, would provide a better platform for this work.

Compared to the original manuscript that the reviewer2 previously reviewed, much of the supplementary information have been moved into the main figure and main text. For example, best images of each cloudinids have been incorporated into Figure 1, so that the readers could get the essence of morphology of each species at a glance. We think there are currently too many supplementary figures to be incorporated into the main text; now even there are two additional supplementary figures to meet the recommendations during this review process. The detailed descriptions of each “true” cloudinid species have been already incorporated in the main text, leaving only the descriptions for the cloudinids not-considered in this study, and the “non-cloudinid cloudinomorphs (or the cloudinid-like tubular organisms as newly emended).” If incorporating these parts into the main text is still considered as necessary, we are happy to do that in the next round of review, but our feeling is that the paper can have most impact if it is relatively short and to-the-point, allowing readers to access the Supplementary materials if they are stimulated to do so.